# Identification of a small-molecule inhibitor that selectively blocks DNA-binding by *Trypanosoma brucei* replication protein A1

Aditi Mukherjee [1], Zakir Hossain[2], Esteban Erben [3,4], Shuai Ma[5], Jun Yong Choi [2,5,6] ✉ & Hee-Sook Kim [1,7] ✉

Replication Protein A (RPA) is a broadly conserved complex comprised of the RPA1, 2 and 3 subunits. RPA protects the exposed single-stranded DNA (ssDNA) during DNA replication and repair. Using structural modeling, we discover an inhibitor, JC-229, that targets RPA1 in *Trypanosoma brucei*, the causative parasite of African trypanosomiasis. The inhibitor is highly toxic to *T. brucei* cells, while mildly toxic to human cells. JC-229 treatment mimics the effects of *Tb*RPA1 depletion, including DNA replication inhibition and DNA damage accumulation. In-vitro ssDNA-binding assays demonstrate that JC-229 inhibits the activity of *Tb*RPA1, but not the human ortholog. Indeed, despite the high sequence identity with *T. cruzi* and *Leishmania* RPA1, JC-229 only impacts the ssDNA-binding activity of *Tb*RPA1. Site-directed mutagenesis confirms that the DNA-Binding Domain A (DBD-A) in *Tb*RPA1 contains a JC-229 binding pocket. Residue Serine 105 determines specific binding and inhibition of *Tb*RPA1 but not *T. cruzi* and *Leishmania* RPA1. Our data suggest a path toward developing and testing highly specific inhibitors for the treatment of African trypanosomiasis.

T*rypanosoma brucei* is a protozoan parasite that causes African trypanosomiasis in humans (Human African Trypanosomiasis, HAT) in sub-Saharan Africa. This parasite also causes nagana, a form of trypanosomiasis found in cattle and other livestock. Transmission of the parasite occurs through biting by infected tsetse flies (*Glossina* spp.)[1]. African trypanosomiasis is one of 20 "neglected tropical diseases" designated by the World Health Organization (WHO). Four drugs and combination therapy have been used for the treatment of HAT: suramin and pentamidine target the early hemolymphatic stage of infection (reviewed in refs. 2, 3); melarsoprol, eflornithine, and a combination of eflornithine and nifurtimox are used for late

meningoencephalitic stage infection in which parasites invade the central nervous system (CNS) after crossing the blood–brain barrier[2,3]. However, melarsoprol is highly toxic and eflornithine is difficult to administer and costly, and many, including melarsoprol, suramin, and pentamidine have problems with drug resistance[2,3]. There are no good treatment options for nagana. New therapeutics are thus urgently needed.

In recent years, several attempts have been made to develop new lines of therapeutics (reviewed in refs. 3, 4). Target-based high-throughput screening (HTS) of a compound library identified *T. brucei* *N*-myristoyltransferase inhibitors[5]. *T. brucei* RNA-editing ligase 1 (REL1)

[1]Public Health Research Institute, Rutgers Biomedical Health Sciences, Newark, NJ 07103, USA. [2]Department of Chemistry and Biochemistry, Queens College, New York, NY 11367, USA. [3]Instituto de Investigaciones Biotecnológicas, Universidad Nacional de San Martín (UNSAM) – Consejo Nacional de Investigaciones Científicas y Técnicas (CONICET), San Martín, Provincia de Buenos Aires, Argentina. [4]Escuela de Bio y Nanotecnologías (EByN), Universidad Nacional de San Martín, San Martín, Provincia de Buenos Aires, Argentina. [5]Ph.D. Program in Chemistry, The Graduate Center of the City University of New York, New York, NY 10016, USA. [6]Ph.D. Program in Biochemistry, The Graduate Center of the City University of New York, New York, NY 10016, USA. [7]Department of Microbiology, Biochemistry, and Molecular Genetics, New Jersey Medical School, Rutgers Biomedical Health Sciences, Newark, NJ 07103, USA. ✉e-mail: junyong.choi@qc.cuny.edu; heesook@njms.rutgers.edu

inhibitors have also been identified by virtual screening of a compound library. These inhibitors are predicted to bind the ATP-binding pocket of the *Tb*REL1 protein[6]. Structure-guided docking studies have identified Pteridine Reductase 1 (PTR1) inhibitors of three highly related trypanosomatids, *T. brucei*, *T. cruzi* (the causative parasite of Chagas disease), and *Leishmania major* (parasite causing leishmaniasis)[7–9]. Phenotypic screening approaches have also identified inhibitors. Fexinidazole, the first all-orally administered drug targeting trypanosomatids, was recently approved by the US Food and Drug Administration (FDA). Several inhibitors are now in preclinical and clinical trials[10].

Another approach that has been applied is compound repurposing or repositioning. This method is less time consuming and more cost effective than HTS of large-scale compound libraries. In fact, many drugs currently used for leishmaniasis and trypanosomiasis are repositioned drugs, including amphotericin B (anti-fungal to anti-leishmaniasis), miltefosine (anti-cancer to anti-leishmaniasis), and nifurtimox (anti-Chagas disease to anti-trypanosomiasis, HAT)[11]. Although fexinidazole is currently in use and other promising candidates are in phase II/III clinical trials, diversity in treatment options would benefit disease control and management of drug resistance[10].

DNA replication proteins regulate genome replication and have additional functions during the replication stress checkpoint and DNA damage response (DDR) (reviewed in refs. 12, 13). The DDR pathways have three components—sensors that recognize damaged DNA; transducers for signal transduction; and effectors that induce cell-cycle arrest, cell death, and/or damage repair[12,13]. The DDR pathways are activated by the recruitment and activation of ataxia-telangiectasia-mutated (ATM) and ATM-Rad3 related (ATR) kinases followed by protein phosphorylation cascades[12,13]. Replication Protein A (RPA) is a eukaryotic ssDNA-binding protein whose function is essential in DNA replication, repair, and recombination[14]. The RPA complex is composed of three subunits, RPA1, RPA2, and RPA3[14]. RPA also acts as a DDR sensor by binding ssDNA regions exposed due to replication fork instability or double-strand breaks (DSBs)[14]. Damaged DNA, marked by RPA-coated ssDNA, recruits a complex of ATR kinase and ATR-Interacting Protein (ATRIP) by the interaction between the N-terminal domain of RPA1 (RPA1-N) and ATRIP[15]. ATR phosphorylates the N-terminal domains of RPA1 and RPA2[16]. RPA1-N is also important for the loading of the 9-1-1 (RAD9-RAD1-HUS1) complex, which recruits TopBP1 (Topoisomerase 2 Binding Protein 1), the activator of ATR[17–19]. CHK1 kinase, phosphorylated by ATR, then coordinates the cell-cycle checkpoint response[18]. Thus, ssDNA binding by RPA plays a vital role in the DNA damage response by recruiting DDR factors to the sites of damage and activating the ATR-mediated cell-cycle checkpoint. Many proteins involved in DDR have been targeted for cancer therapy and some of these proteins[11,12,20] are conserved in *T. brucei*, including ATR, ATM, and RPA[21–24]. Hence, repositioning existing compounds that target these proteins might provide opportunities for the development of anti-trypanosomal therapeutics.

Eukaryotic RPA subunits, including the human ortholog, contain six oligosaccharide/oligonucleotide-binding (OB) folds: four in the largest subunit RPA1 and one each in RPA2 and RPA3[14]. The OB-fold is found in many proteins with ssDNA-binding functions[25,26]. RPA1 has domains called DBD (DNA-binding domain): DBD-F, A, B, and C in tandem. Each of these DBDs contains an OB-fold structure. While DBD-F also contains an OB-fold structure, it is involved in the mediation of protein–protein interactions (PPIs). The central region of the RPA1, DBD-A, and B (DBD-AB), is involved in dynamic interaction with ssDNA and DBD-C is required for the interaction of RPA1 with RPA2[15,26–28].

Several laboratories have developed RPA1 inhibitors for cancer therapy by targeting specific functions of RPA1[29–38]. For example, the Turchi group identified TDRL-505 and its derivatives, which inhibit the ssDNA-binding activity of DBD-AB[29,30]. TDRL-505 treatment inhibited the proliferation of human non-small cell lung cancer H460 cells

($IC_{50} = 30.8\,\mu M$)[29]. The Oakley laboratory screened compounds that inhibit the interaction of RPA1 DBD-F with RAD9 and identified two inhibitors, HAMNO (known as NSC111847) and NSC15520[31,32]. The Fesik laboratory utilized NMR spectroscopy to screen libraries of compounds that bind the DBD-F of RPA1 and identified a series of compounds that bind to two adjacent sites within the DBD-F[34,35,37]. Some of these compounds blocked the interaction of human RPA1 DBD-F with ATRIP peptides at sub-micromolar levels[34,35,37,38], which would potentially disrupt the DDR pathways and promote cell death.

All three subunits of RPA are present in trypanosomatids, including *T. brucei*, *T. cruzi*, and *Leishmania* spp. Trypanosomatid RPA1 sequences and domain structures are mostly conserved as in mammals, except that trypanosome RPA1 proteins do not have the DBD-F (Fig. 1a and Supplementary Fig. 1a)[39–42]. The molecular functions of RPA in these parasites have not been thoroughly explored and only limited information is available to date. In *T. cruzi*, electrophoretic mobility shift assay (EMSA) using recombinant DBD-A, B, or C demonstrated that *Tc*RPA1 DBD-A has the strongest ssDNA-binding activity[42]. *Tc*RPA1 and *Tc*RPA2 form nuclear repair foci[42]. Interestingly, *Tc*RPA relocalizes to the cytoplasm during the non-proliferative metacyclic and bloodstream trypomastigote stages[42,43]. Cytoplasmic localization of RPA has not been reported in mammals or yeasts. This parasite may use nuclear export of RPA to control proliferative and non-proliferative phases during life-cycle specific differentiation. RPA1 has been shown to bind telomeric ssDNA both in *T. cruzi* and *Leishmania*[44,45], while in humans and yeast, telomeric ssDNA binding and protection are linked to a separate complex called CST (CTC1-STN1-TEN1)[46]. In *Leishmania*, RPA1 also binds RNA[47]. *T. brucei* RPA is the least studied among these three trypanosomatids. *T. brucei* RPA1 also forms nuclear foci in response to DSBs[48,49]. Although available data are limited, unique features of RPA1 in these parasites, such as its nuclear export during differentiation and roles in telomere protection, offer opportunities for selective therapeutic development[39–42,44,45].

Here we demonstrate that *T. brucei* replication proteins are potential therapeutic targets for the treatment of African trypanosomiasis. We first characterized phenotypes associated with the depletion of *Tb*RPA1 protein and examined the effects of existing or repositioned *Hs*RPA1 PPI inhibitors on *T. brucei* cell growth. We discover that JC-229, an analog of an *Hs*RPA1 inhibitor, possesses specific inhibitory effects on *Tb*RPA1 protein, but not on human RPA1 in vitro. Site-directed mutagenesis driven by molecular modeling confirms that ssDNA and JC-229 share a binding site in DBD-A of *Tb*RPA1. In particular, the S105T mutation completely abolishes the inhibitory effect of JC-229 without altering the ssDNA-binding capacity of *Tb*RPA1 in vitro and confers resistance to JC-229 in trypanosome cells. These data point toward a therapeutic that targets *Tb*RPA1 while having minimal cytotoxic effects on human cells.

## Results

### Identification of JC-229 as an inhibitor of *T. brucei* cell growth and its potential interaction with *Tb*RPA1

RPA1 has three ssDNA-binding domains (DBD-A, DBD-B, and DBD-C) and an N-terminal DBD-F for PPI function in eukaryotic model systems[14]. However, *T. brucei* RPA1 lacks the N-terminal domain, although the three ssDNA-binding domains are present (Fig. 1a and Supplementary Fig. 1a). Inhibitors specifically targeting the PPI or the ssDNA-binding activity of human RPA1 are being developed as anti-cancer therapeutic candidates[29–38]. Given that all four domains, each have an OB-fold structure, a known motif of ssDNA-binding, we hypothesized that if some of these inhibitors bind and act at the OB-fold structure, those human RPA1 inhibitors may also be able to bind the OB-fold of *Tb*RPA1 and inhibit *Tb*RPA1 function. Since no NMR or crystallographic *Tb*RPA1 structures are currently available, we generated a 3D structural model of *Tb*RPA1 by applying homology modeling in the SWISS-MODEL server and compared with the structure of *Hs*RPA1. *Hs*RPA1 and *Tb*RPA1

have 54% sequence identity in DBD-A and 34% in DBD-B (Supplementary Fig. 1a). As shown in Fig. 1b (left), the structure of *Hs*RPA1 DBD-F (in short, *Hs*DBD-F, blue) is overlaid to that of *Tb*RPA1 DBD-A (in short, *Tb*DBD-A, red), and it shows that both DBD-F and DBD-A domains have similar tertiary structures with aligned five beta-sheet strands where DNA binds. *Hs*DBD-F harbors a binding pocket for ATRIP peptide-37 (green) (PDB code: 4NB3) (Fig. 1b, middle) and *Tb*RPA1 DBD-A binds ssDNA (Fig. 1b, right). We found that *Hs*DBD-F does not align well with *Tb*RPA1 DBD-B (in short, *Tb*DBD-B), unlike *Tb*DBD-A, as shown in Supplementary Fig. 1b, c. Structures from AlphaFold are very similar to homology model structures (Supplementary Fig. 1d, e). ATRIP peptide-37 is shown to interact with *Hs*DBD-F and *Hs*DBD-AB with similar binding affinities ($K_d = 7.4\,\mu M$ and $12\,\mu M$, respectively)[36], which also supports our hypothesis that *Hs*DBD-F inhibitors could bind *Tb*DBD-A and block the interaction between *Tb*RPA1 and ssDNA.

As proof-of-principle, we synthesized and tested five compounds, including four (JC-230–233) known *Hs*DBD-F inhibitors[35,37], and one analog (JC-229) of compound **3** or **4** in ref. 20 (Supplementary Fig. 2a, b

and Fig. 1d). JC-229 was validated by Nuclear Magnetic Resonance (NMR) (Supplementary Fig. 3). We found that JC-229 treatment abolished parasite proliferation (Fig. 1e). We then titrated JC-229 concentration to obtain a Half Maximal Effective Concentration ($EC_{50}$) with serial dilutions of JC-229 (from 0.5 to 50 μM) and determined viability using the AlamarBlue assay (Fig. 1f). JC-229 was toxic to *T. brucei* cells with an $EC_{50}$ of 6.6 μM (±0.4). With JC-230−233, the known *Hs*DBD-F inhibitors[35,37], we saw no significant toxic effect on trypanosome cell growth (Supplementary Fig. 2c−f).

Induced-fit docking with the homology model showed that JC-229 occupies the expected ssDNA-binding pocket of *Tb*RPA1 DBD-A (Fig. 1c). Particularly, the sulfonamide and amide units of JC-229 form hydrogen bond interactions with Q87 and N92/R60, respectively, and two dichlorophenyl rings have π · π stacking interactions with F64 and F95 of *Tb*RPA1 DBD-A (Fig. 1c). We also examined TDRL-505, a known inhibitor of *Hs*RPA1 ssDNA-binding activity[29], but did not observe any effect on *T. brucei* proliferation even at 60 μM concentration (Supplementary Fig. 2g).

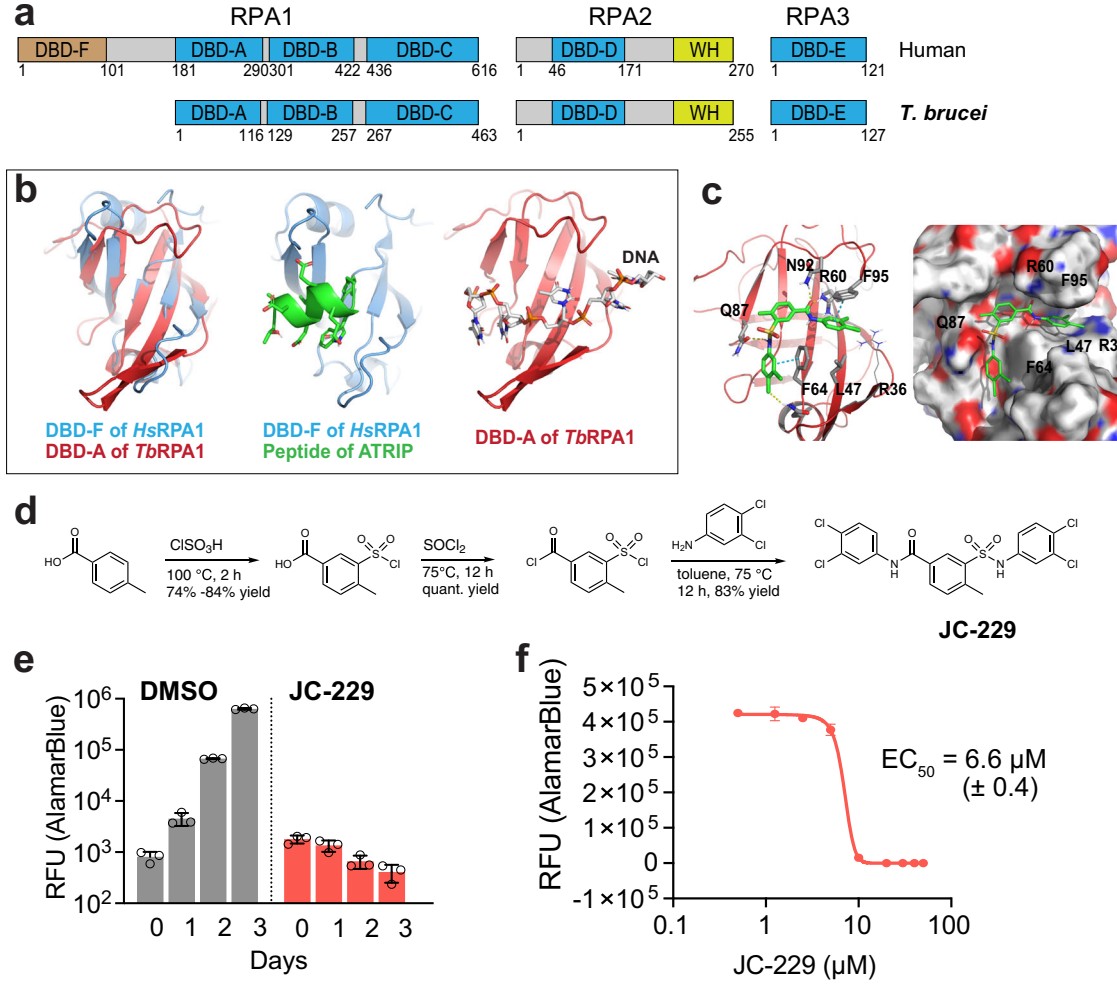

**Fig. 1 | Structural model of *Tb*RPA1 DBD-A in complex with JC-229 and inhibition of *T. brucei* cell proliferation by JC-229. a** Schematic diagram of the three subunits that comprise human and *T. brucei* RPA complexes. Positions of amino acid residues defining each domain are shown. DBD DNA-binding domain and WH winged helix. **b** Left: The structure alignment of *Hs*DBD-F (light blue) to *Tb*RPA1 DBD-A (red) generated by Pymol software (SWISS-MODEL). Middle: X-ray co-crystal structure of *Hs*DBD-F (light blue) in complex with ATRIP derived peptide (green) (PDB code: 4NB3). Right: A 3D homology model of *Tb*RPA1 DBD-A (red) interacting with ssDNA (SWISS-MODEL). **c** Induced-fit docking model showing the binding position of JC-229 in the ssDNA-binding pocket of *Tb*RPA1 DBD-A (ribbon (left) and

sphere (right) models shown). **d** Synthesis scheme for JC-229. **e** Inhibition of *T. brucei* cell growth by JC-229. Wild-type (WT) *T. brucei* cells were treated with 50 μM JC-229 for 3 days and viability was determined using the AlamarBlue assay. Three biological replicates were used for each measurement ($n = 3$). Error bars indicate mean ± SD. **f** Dose-dependent growth inhibition of *T. brucei* by JC-229. *T. brucei* cells were treated with increasing concentrations of JC-229 (0.5 to 50 μM) for 72 h and viability was determined with the AlamarBlue assay. A standard 4-Parameter Logistic (4-PL) curve from GraphPad Prism software generated the $EC_{50}$ value. All growth experiments were performed in triplicate using biological replicates ($n = 3$). Error bars indicate mean ± SD. Source data are provided as a Source Data file.

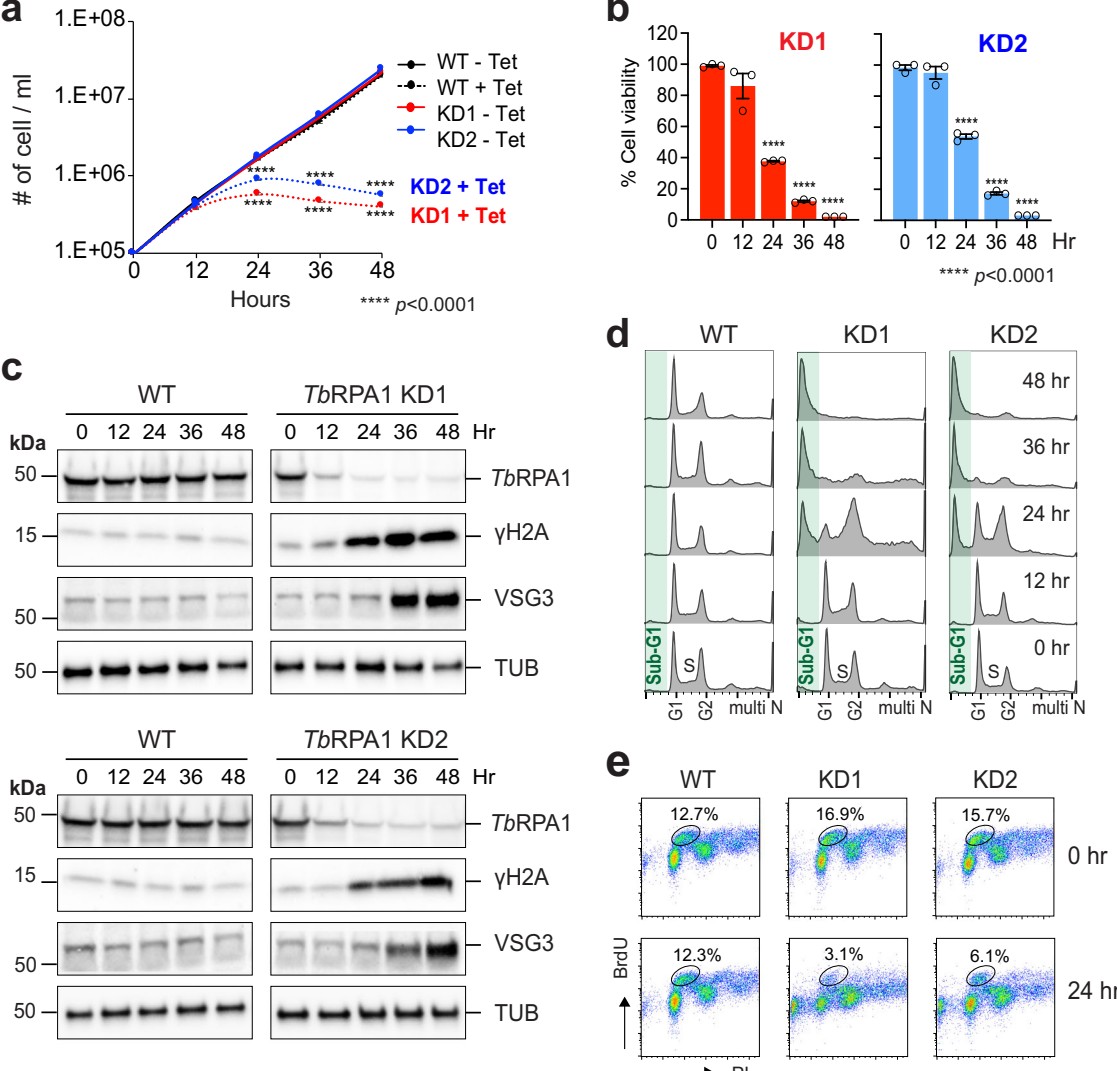

**Fig. 2 | Depletion of *Tb*RPA1 induces cellular lethality, abnormalities in cell-cycle progression, and DNA replication. a** *Tb*RPA1 is essential for *T. brucei* cell viability. *Tb*RPA1 RNAi depletion was induced with the addition of tetracycline and cell growth was monitored by counting cells every 12 for 48 h. WT was used as a control. Three technical replicates were used for each measurement ($n = 3$). Error bars indicate mean ± SD. An unpaired two-sided Student's *t*-test was performed, and statistically significant results were indicated with ****$p < 0.0001$. **b** Viability of *Tb*RPA1-depleted trypanosome cells. Cells from each time point in Fig. 2a were examined for viability using the AlamarBlue assay. Percent viability compared to non-depleted cells was determined and plotted. Three technical replicates were used for each measurement ($n = 3$). Error bars indicate mean ± SD. An unpaired two-

sided Student's *t*-test was performed, and statistically significant results were indicated with ****$p < 0.0001$. **c** Immunoblot controls presenting the changes in protein levels for *Tb*RPA1, γH2A, VSG3, and Tubulin at each time point after depletion. Tubulin serves as a loading control. Three independent experiments were performed with similar results. **d** Cell-cycle profiles of *Tb*RPA1-depleted cells. Fixed cells were stained with PI and analyzed by flow cytometry. **e** DNA synthesis assay by BrdU pulse labeling. WT and *Tb*RPA1 KD cells were pulse-labeled with 500 μM BrdU and fixed. Fixed cells were then stained with PI (bulk DNA) and anti-BrdU-Alexa 488 antibody (newly synthesized DNA) and analyzed by flow cytometry. Source data are provided as a Source Data file.

## Depletion of *T. brucei* RPA1 leads to cell lethality and inhibition of DNA replication

To see whether JC-229 impacts *Tb*RPA1 function, we first examined cellular phenotypes following *Tb*RPA1-depletion. *Tb*RPA1 knockdown (KD) was achieved by the expression of dsRNA targeting the *Tb*RPA1 transcript under tetracycline control (Tet-On system)[50]. Stagnation of growth was observed after 24 h of RNAi induction in two independent KD cell lines (KD1 and KD2). The number of cells further decreased after 36 and 48 h (Fig. 2a). To determine the viability of KD cells, we performed the AlamarBlue assay (Fig. 2b). The majority of cells detected after 36 h were dead or dying with 12% (KD1) and 17% (KD2) viability. Only 2% (KD1) and 3% (KD2) of cells were living after 48 h. These data confirm that *Tb*RPA1 is essential for cell viability. Depletion

of *Tb*RPA1 protein in KD1 and KD2 cell lines was confirmed by immunoblotting (Fig. 2c and Supplementary Fig. 4a, b).

*Tb*RPA1 forms nuclear repair foci in response to DSBs or replication stress[48,49]. We predicted that DNA damage would accumulate in the absence of *Tb*RPA1. As we expected, the levels of phosphorylated H2A (γH2A, the equivalent of γH2AX in mammals), which is a marker of DSB[51], increased after *Tb*RPA1 depletion (Fig. 2c and Supplementary Fig. 4c, d).

The cell surface of *T. brucei* is densely coated with a single species of variant surface glycoprotein (VSG). Although the *T. brucei* genome carries over 2000 VSG alleles, the parasite expresses only one VSG allele per cell at any given time and suppresses the expression of the other alleles (VSG silencing). *T. brucei* has mechanisms to switch the expressed VSG, which removes the old VSG coat and presents a new VSG on

the surface. This VSG switching mechanism, known as antigenic variation, allows the parasite to escape host immune recognition and killing (reviewed in refs. [52], [53]). VSG switching events can be triggered by a DSB[54,55]. Because DSBs accumulate in the absence of *Tb*RPA1, the VSG switching rate may increase in KD cells, which allows the expression of transcriptionally silent VSGs and makes them detectable by immunoblotting. De-repression of silenced VSG alleles could also increase the levels of silent VSG proteins, which can be detected by immunoblotting as well. We observed the increase in the levels of VSG3 proteins upon *Tb*RPA1 depletion (Fig. 2c and Supplementary Fig. 4e, f), suggesting that *Tb*RPA1 has roles in VSG expression and/or switching control.

RPA is required for DNA replication initiation and elongation[14]. Thus, we anticipated to see defects in DNA replication and cell-cycle progression in *Tb*RPA1-depleted cells. *Tb*RPA1-depleted cells were fixed, stained with Propidium Iodide (PI) to stain bulk DNA and analyzed by flow cytometry. Two clear aberrations were observed in the absence of *Tb*RPA1: the accumulation of G2-phase cells followed by an increase in sub-G1 cells (<2C DNA content) (Fig. 2d and Supplementary Fig. 20). At 48 h after KD induction, ~60% cells showed <2C DNA content. In the absence of *Tb*RPA1, cells may arrest at the G2/M phase initially and die eventually, thus generating a sub-G1 population. DNA synthesis was also compromised in the absence of *Tb*RPA1, as expected (Fig. 2e and Supplementary Fig. 4g, 21). The number of S phase cells incorporating BrdU, a dT analog that is incorporated into newly synthesized DNA during the S phase, was reduced after the depletion of *Tb*RPA1.

## JC-229 treatment resembles *Tb*RPA1 depletion and has higher toxicity against *T. brucei* than to human cells

If JC-229 indeed binds and inhibits the functions of *Tb*RPA1, as predicted from our structural model, JC-229-treated trypanosome cells should show similar phenotypes to *Tb*RPA1-depleted cells. *T. brucei* cells treated with 10 μM JC-229 were monitored for the cell-cycle profile, efficiency of DNA synthesis, and the levels of *Tb*RPA1, γH2A, VSG3, and Tubulin proteins (Fig. 3a–d).

Abnormalities in cell-cycle progression were observed at 8 and 16 h after JC-229 treatment with a reduction in the S phase population (Fig. 3a and Supplementary Fig. 22a). In particular, the early S phase population (indicated with red arrowheads) was diminished at the 16 and 24-h time points. This could be due to the role of RPA during the initiation of DNA replication. We interpreted that replication origins cannot be activated in the absence of *Tb*RPA1. Consequently, cells are unable to enter the S phase, thus reducing the early S phase population. At the 32-h time point, a drastic increase in the sub-G1 population was observed (~45%) and almost all cells (~97%) lost the bulk of their DNA by 48 h. JC-229 completely blocked new DNA synthesis after 8 h of treatment (Fig. 3b and Supplementary Fig. 22b). Importantly, JC-229 treatment did not change *Tb*RPA1 protein levels (Fig. 3c, d), indicating that phenotypes observed in JC-229 treated cells occur due to the loss of *Tb*RPA1 functions, such as ssDNA-binding activity, rather than the loss of protein itself. The levels of γH2A and VSG3 increased after JC-229 treatment (Fig. 3c, d). These data suggest that JC-229 likely targets an essential function(s) of the *Tb*RPA1 protein. The microscopic analysis confirmed that JC-229 treatment leads to cell death (Fig. 3e and Supplementary Fig. 5). Trypanosome cells started losing their normal morphology (an elongated shape with flagellum attached to the body) after 16 h of JC-229 treatment. After 24 h, the majority of cells were dying, showing a rounded morphology that is typical of dying *T. brucei* cells (Fig. 3e). After 32 and 48 h, unusually small cells that had <2 C DNA were observed frequently (Supplementary Fig. 5).

JC-229 is an analog of compounds that inhibit the interaction of the *Hs*DBD-F with ATRIP-derived peptides and ATRIP peptide-37 can interact with *Hs*DBD-F and *Hs*DBD-AB[33,36,38]. Thus, it is possible that JC-229 could inhibit the functions of *Hs*DBD-F or *Hs*DBD-AB, resulting in growth inhibition of human cells. Therefore, we investigated whether JC-229 inhibition is *T. brucei* selective or not. We determined the toxicity of JC-229 to *T. brucei*, HeLa, and HEK293 cells, using the AlamarBlue assay. JC-229 kills *T. brucei* cells more effectively than human cells (Fig. 3f and Supplementary Table 1). JC-229 exhibited $EC_{50}$ values of 6.5 μM (±0.4), 18.4 μM (±2.2), and 32.1 μM (±2.9) for *T. brucei*, HEK293, and HeLa, respectively: 100% inhibition of HEK293 and HeLa was not observed even at 50 μM, the maximum concentration of JC-229. In addition, HEK293 and HeLa cells are barely inhibited by JC-229 at 6.5 μM ($EC_{50}$ to *T. brucei*).

## JC-229 specifically inhibits the ssDNA-binding activity of *T. brucei* RPA in vitro

Measuring binding kinetics between the purified RPA protein and isotope or fluorescent dye-labeled oligonucleotides is a well-established method for studying RPA's biochemical functions[31,56,57]. To confirm the specific inhibition of *Tb*RPA1 by JC-229 in vitro, we performed electrophoretic mobility shift assays (EMSA) and microscale thermophoresis (MST) assays using RPA proteins purified from *E. coli* and a 32-nucleotide ssDNA substrate (oligo $dT_{32}$) labeled with IRDye800 or Cy5 at the 5´ end.

It is known that RPA subunits are not soluble by themselves, but they fold properly and form a stable complex of RPA when expressed together[58]. Consistently, we were unable to obtain soluble *Tb*RPA1 protein. It has been shown that deleting DBD-C can increase the solubility of truncated *Hs*RPA1 protein[27], since this domain engages in complex formation by interacting with RPA2. Without RPA2, the DBD-C may misfold, making full-length RPA1 insoluble. We were able to express soluble 6xHis-tagged *Tb*RPA1 DBD-AB (lacks DBD-C) from *E. coli*. We purified *Tb*RPA1 DBD-AB protein (in short, *Tb*DBD-AB) using Ni⁺⁺-nitrilotriacetic acid (Ni-NTA) purification (Supplementary Fig. 6a and Fig. 4i). The purified *Tb*DBD-AB protein (amino acids 1–257) was used to study the ssDNA-binding properties of *Tb*RPA1 and the inhibitory effect of JC-229.

Increasing concentrations of *Tb*DBD-AB proteins (0.05–100 nM) were incubated with a 5´IRDye800-labeled-$dT_{32}$ probe. Unbound and protein-bound probes (RPA-ssDNA complex) were separated in a native PAGE gel and visualized with the LI-COR imaging system. Signals from the ssDNA-protein complex become stronger with increasing protein concentrations (Fig. 4a). To see if JC-229 inhibits the ssDNA-binding activity of *Tb*DBD-AB, we pre-incubated recombinant *Tb*DBD-AB protein with 2-fold serial dilutions of JC-229 (0.625–80 μM) and then added 5´IRDye800-labeled oligos. The ssDNA binding of *Tb*DBD-AB was inhibited by JC-229 in a dose-dependent manner (Fig. 4b). We then performed the same assay using purified *Hs*DBD-AB (Supplementary Fig. 6b and Fig. 4i), to see whether JC-229 inhibits the *T. brucei* protein, but not the human DBD-AB. Recombinant *Hs*DBD-AB protein bound ssDNA in a concentration-dependent manner (Fig. 4c). However, even the highest concentration of JC-229 (80 μM) did not inhibit the ssDNA-binding activity of the human RPA1 protein (Fig. 4d). We conclude that JC-229 targets *Tb*RPA1 protein and importantly, it exhibits specificity toward *Tb*RPA1.

The RPA functions as a complex, rather than as an individual subunit. The RPA complex has multiple DBDs with varying levels of activities, and while all DBDs each have an OB-fold[26], the major dynamic ssDNA-binding activity is present in DBD-A and DBD-B[27,28,59]. To examine the ssDNA-binding properties in this more natural context, we performed EMSA using recombinant RPA complexes. We engineered a vector for the simultaneous expression of 6xHis-tagged RPA1, RPA2, and RPA3 subunits in *E. coli* (only the RPA1 subunit has a 6xHis-tag). Simultaneous expression of all three subunits in the same cell facilitates protein folding and complex formation[58]. A full RPA complex can be pulled down via the 6xHis-tagged RPA1 subunit with Ni-NTA resin. We purified both *Tb*RPA and *Hs*RPA complexes (Fig. 4j and Supplementary Fig. 6c, d) and used these proteins in EMSA assays as above. We confirmed the presence of all subunits in purified fractions by mass spectrometry (Supplementary Fig. 7) and

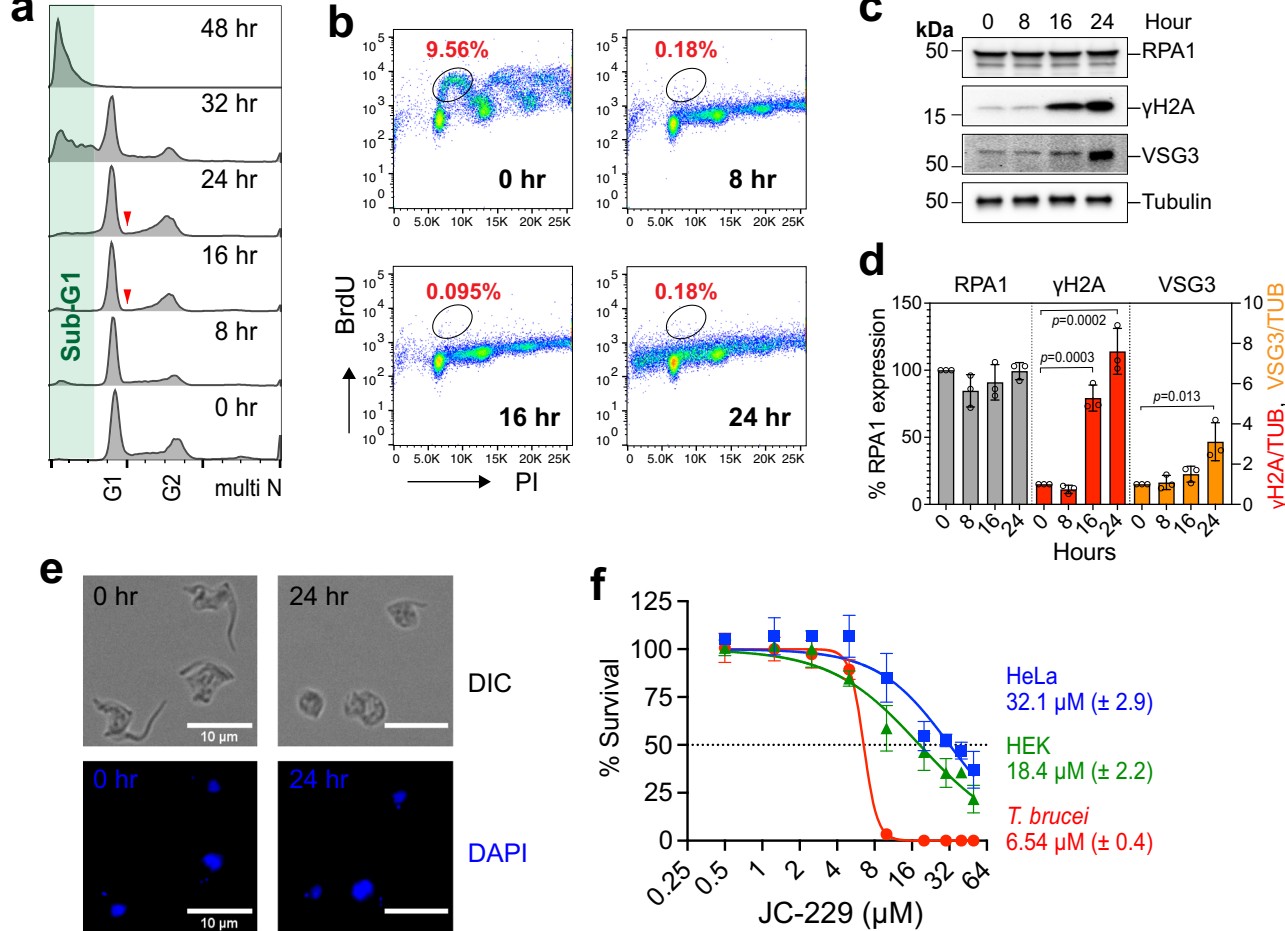

**Fig. 3 | JC-229 treatment mimics *Tb*RPA1 depletion and is more toxic to *T. brucei* than human cells. a** Cell-cycle profiles of JC-229-treated *T. brucei* cells. WT *T. brucei* cells were treated with 10 μM of JC-229 and collected at indicated time points. Cells were fixed, stained with PI, and analyzed by flow cytometry. **b** DNA synthesis assay by BrdU pulse labeling. *T. brucei* cells treated with 10 μM JC-229 for 0, 8, 16, and 24 h were pulse-labeled with BrdU as shown in Fig. 2e. Cells were fixed, stained with PI and anti-BrdU-Alexa 488 antibody, and then analyzed by flow cytometry. **c** Immunoblots showing the levels of *Tb*RPA1, γH2A, VSG3, and Tubulin proteins in cells treated with JC-229 for indicated hours. Tubulin serves as a loading control. **d** Quantification of protein levels in immunoblots. Signal intensities of protein bands were quantified using ImageJ software, normalized to those of Tubulin bands. Three independent experiments were performed ($n = 3$) and fold changes relative to the untreated samples were plotted. Error bars indicate mean ± SD. An unpaired two-sided Student's *t*-test was performed. **e** Microscopic analysis of JC-229 treated *T. brucei* cells. Cells collected at indicated time points were fixed in 1% paraformaldehyde for 10 min and stained with DAPI. Images were obtained using an EVOS M5000 microscope. Scale bar, 10 μm. Twenty images were taken for each time point with similar results. **f** Cytotoxicity of JC-229 on *T. brucei* and two human cell lines (HeLa and HEK293). A dose-response curve was obtained from cells treated with increasing concentrations of JC-229 (0.5, 1.25, 2.5, 5, 10, 20, 30, 40, and 50 μM) for 72 h. Three biological replicates were used for each measurement ($n = 3$) and EC50 values were obtained from GraphPad Prism software (inhibitor vs. normalized response with variable slopes). Error bars indicate mean ± SD. Source data are provided as a Source Data file.

oligomerization of the *Tb*RPA complex by size exclusion chromatography (SEC) and mass photometry (MP) (Supplementary Fig. 8). The formation of ssDNA-protein complexes was dependent on the protein concentration for both *Tb*RPA and *Hs*RPA (Fig. 4e, g). JC-229 inhibited *Tb*RPA complex binding to ssDNA, while causing no effect on binding of the *Hs*RPA complex to ssDNA (Fig. 4f, h). The data confirm the specificity of JC-229 against *T. brucei* RPA. All EMSA assays were performed in triplicate, and signals were quantified and plotted as shown in Supplementary Fig. 9.

To accurately measure the binding constant and IC50 values, we performed MST assays using the 5′Cy5-labeled-dT32 ssDNA probe and used the proteins in the EMSA experiments above. We observed essentially the same trend in binding and inhibition (Fig. 5 and Supplementary Tables 2, 3), as in the EMSA experiments. We obtained the following $K_d$ values for ssDNA binding: 24.6 nM for *Tb*DBD-AB, 13.2 nM for *Tb*RPA complex, 48.0 nM for *Hs*DBD-AB, and 15.8 nM for *Hs*RPA complex (Fig. 5a, c, e, g). JC-229 strongly inhibited the binding activity of *Tb*DBD-AB and the *Tb*RPA complex, exhibiting IC50 values of 228

and 24.5 nM (Fig. 5b, d), respectively, while having no inhibitory effect on human RPA proteins (Fig. 5f, h). *Tb*RPA1 DBD-AB has a weaker binding affinity to ssDNA compared to the *Tb*RPA complex (~2-fold), but a much higher concentration of JC-229 is needed to inhibit the activity of *Tb*DBD-AB than the *Tb*RPA complex (228 vs. 24.5 nM, ~9-fold). Approximately three DBD-AB molecules can bind the 32 nt oligo probe, while only one RPA complex can bind the same oligo since other DBDs (DBD-C, D & E) in the RPA complex also engage in ssDNA binding[14,60]. Therefore, a higher concentration of JC-229 may be needed to inhibit the interaction of *Tb*DBD-AB with oligo dT32. Or alternatively, conformations of *Tb*DBD-AB in truncated form and the natural RPA complex are different and JC-229 may have easier access to *Tb*DBD-A in the full complex than in the DBD-AB fragment.

If more JC-229 is required to inhibit *Tb*DBD-AB than the *Tb*RPA complex because dT32 can harbor more molecules of *Tb*DBD-AB, we anticipated that less JC-229 is needed to block the interaction of *Tb*DBD-AB with shorter oligo substrates. We performed EMSA and MST experiments using oligo dT20 and dT12 with purified *Tb*DBD-AB

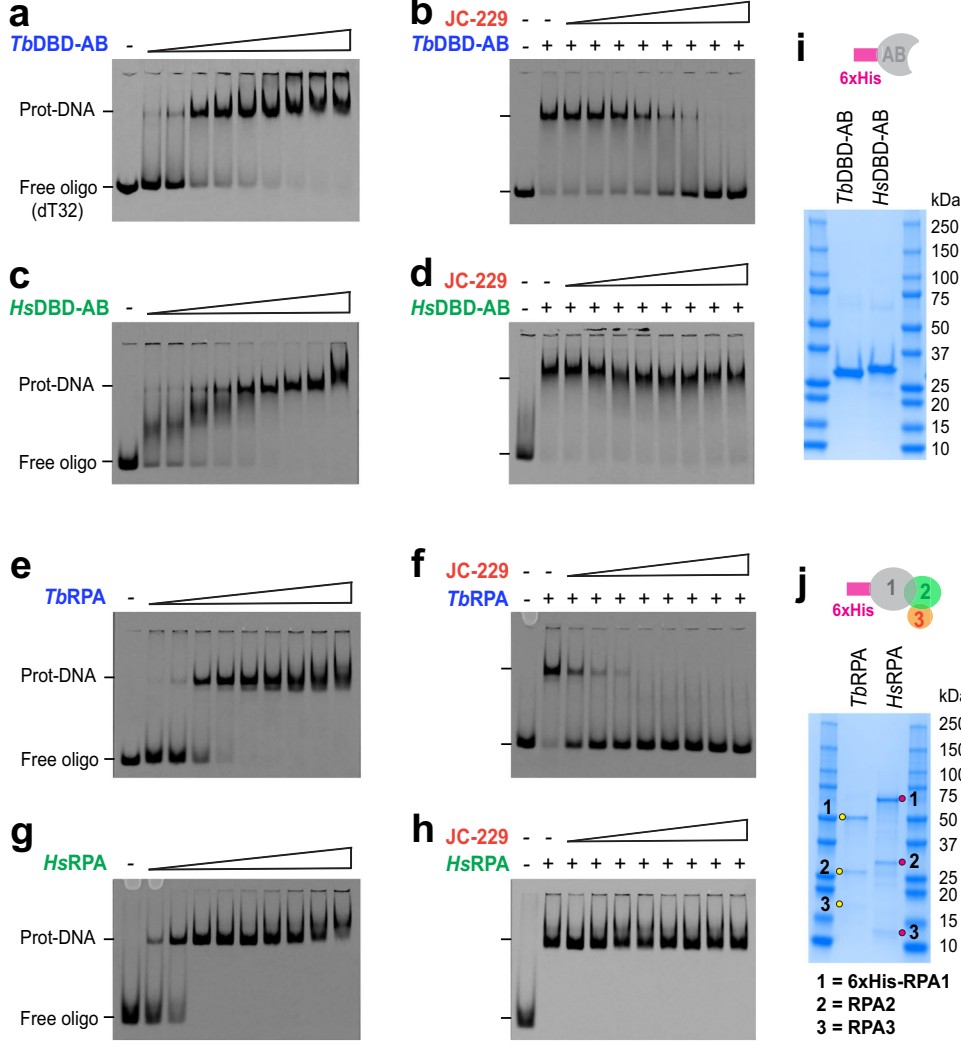

**Fig. 4 | JC-229 specifically inhibits the ssDNA-binding activity of *Tb*RPA.** EMSA used to measure the ssDNA-binding activity of recombinant RPA proteins (gel images, **a**, **c**, **e**, **g**) and inhibition of this binding activity by JC-229 (gel images, **b**, **d**, **f**, **h**) are shown. **a** ssDNA-binding activity of *Tb*RPA1 DBD-AB. Increasing concentrations (0.05, 0.2, 1.6, 3.2, 6.25, 12.5, 25, 50, and 100 nM) of *Tb*RPA1 DBD-AB protein were incubated with 2 nM oligo dT$_{32}$ labeled with 5´IRDye (5´IRDye800-dT$_{32}$) and the protein–ssDNA complex was analyzed by EMSA. **b** Inhibition of the ssDNA-binding activity of *Tb*RPA1 DBD-AB by JC-229. Serial dilutions (0.625, 1.25, 2.5, 5, 10, 20, 40, and 80 µM) of JC-229 were pre-incubated with 12.5 nM *Tb*RPA1 DBD-AB protein, then incubated with 5´IRDye800-dT$_{32}$. Inhibition of binding was visualized by EMSA. **c**–**h** The same binding assay and JC-229 inhibition assay were performed with purified *Hs*RPA1 DBD-AB protein (**c**, **d**), with the *Tb*RPA complex (**e**, **f**) and *Hs*RPA complex (**g**, **h**). **i**, **j** SDS-PAGE showing purified fractions used for EMSA assays. Three independent EMSA experiments were performed for Fig. 4a–h with similar results. Source data are provided as a Source Data file.

and *Tb*RPA complex (Supplementary Figs. 10, 11 and Supplementary Tables 2, 3). *Tb*DBD-AB bound dT$_{20}$ and dT$_{32}$ with similar kinetics ($K_d$: 30.5 nM for dT$_{20}$ vs. 24.6 nM for dT$_{32}$). Surprisingly, inhibition of the binding between dT$_{20}$ and *Tb*DBD-AB required more JC-229 (IC$_{50}$: 881 nM for dT$_{20}$ vs. 228 nM for dT$_{32}$). These data suggest that *Tb*DBD-AB and dT$_{20}$ complex is more stable than *Tb*DBD-AB and dT$_{32}$ complex. With the *Tb*RPA complex, similar $K_d$ and IC$_{50}$ values were obtained from dT$_{20}$ and dT$_{32}$. $K_d$ values are 15.5 nM for dT$_{32}$ vs. 17.0 nM for dT$_{20}$ and IC$_{50}$ values are 33.9 nM for both dT$_{32}$ and dT$_{20}$. Both *Tb*DBD-AB and *Tb*RPA complex did not bind dT$_{12}$ well. Thus, as small as 20-mer can form a stable complex with both the DBD-AB domain and RPA complex in *T. brucei*.

To ensure that our purified proteins do not have residual *E. coli* SSB protein (S̲ingle-S̲tranded D̲NA-B̲inding protein, the bacterial counterpart of RPA) contamination and also to see whether *Tb*DBD-A has a stronger binding affinity than *Tb*DBD-B as in other eukaryotic model systems, we purified and examined two mutant *Tb*RPA1 DBD-AB proteins, F64A mutant (in DBD-A, F238 in human) and

W188A mutant (in DBD-B, W361 in human) (Supplementary Fig. 12). In human, DBD-AB F238A mutant protein retained ~30% activity compared to WT DBD-AB, while W361A mutant lost 90% activity and F238A W361A double mutant completely lost the binding activity (>500-fold reduction)[59]. These indicate that the presence of either F238 or W361 is sufficient for some binding of *Hs*RPA to ssDNA. Different from human DBD-AB, we observed no binding from *Tb*DBD-AB F64A or W188A single mutant, suggesting that in *T. brucei*, both DBD-A and B are required for a stable binding of DBD-AB with ssDNA. Consistent with this result, *Tb*DBD-A alone (amino acids 1–116) or *Tb*DBD-B alone (amino acids 129–257) did not show much binding activity to both dT$_{32}$ and dT$_{20}$ substrate (Supplementary Fig. 13 and Supplementary Table 2).

While JC-229 has no inhibitory effect on the ssDNA-binding activity of the *Hs*DBD-AB and *Hs*RPA complex, it is possible that JC-229 may inhibit the PPI function of *Hs*RPA1. We purified 6xHis-*Hs*DBD-F (amino acids 1–181), synthesized ATRIP peptide, and examined whether we could detect the interaction of *Hs*DBD-F with ATRIP and

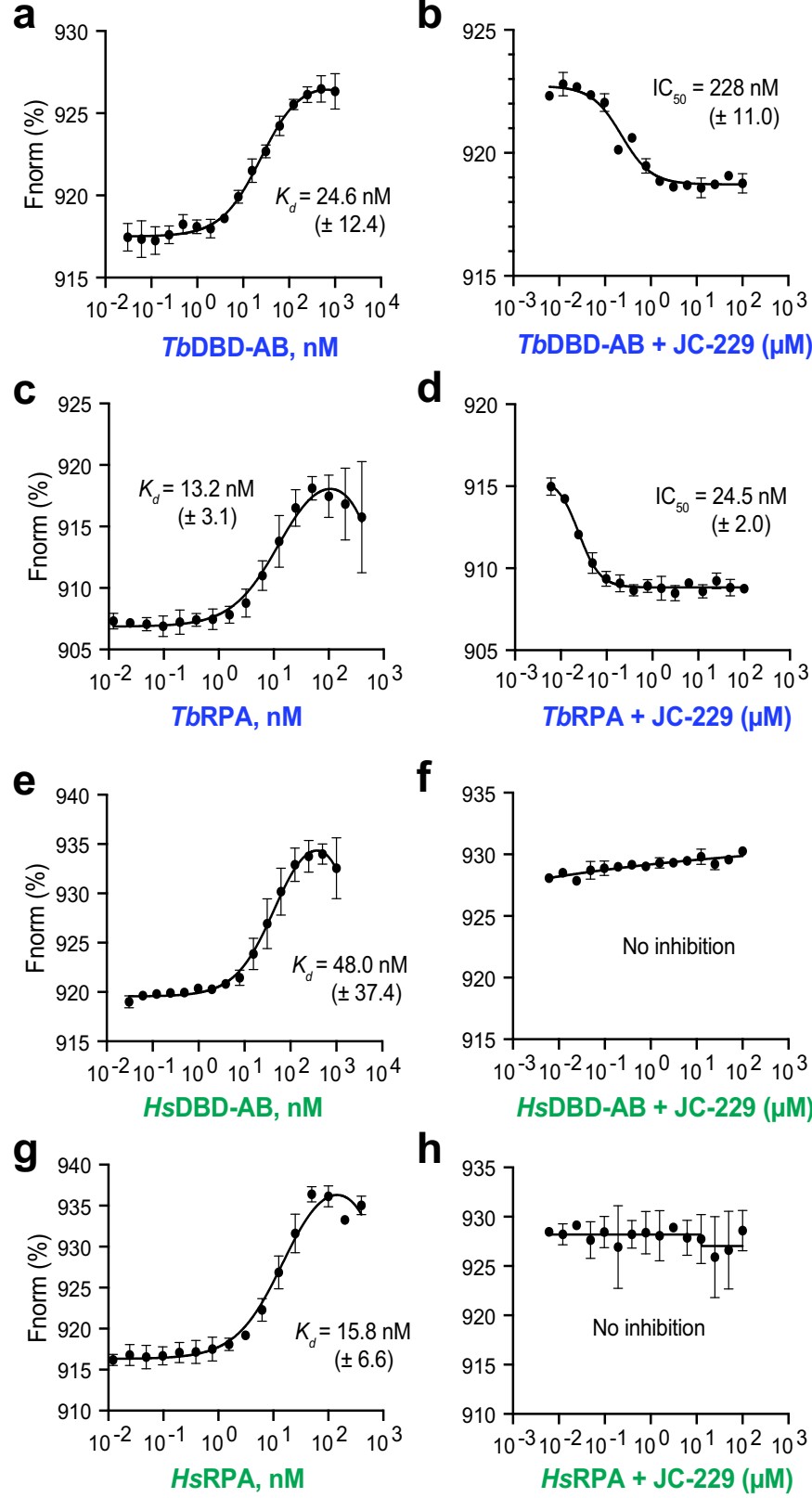

whether JC-229 inhibits their interaction (Supplementary Figs. 14, 15 and Supplementary Table 4). We found that purified *Hs*DBD-F binds the ATRIP peptide with $K_d$ of 34.3 μM and JC-229 inhibits this interaction with $IC_{50}$ of 60.8 μM. Although JC-229 inhibits the function of human RPA1, it does so at a very high concentration. This may partly explain the mild toxicity of JC-229 towards human cells (Fig. 3f).

### *T. brucei* RPA1 DBD-A has the binding pocket for JC-229: Ser 105 in DBD-A is important

Sequence identity between DBD-ABs of *T. brucei* and *T. cruzi* is ~83 and 72% between *T. brucei* and *Leishmania*. This high level of identity suggested that JC-229 may also inhibit the ssDNA-binding activity of *T. cruzi* and *Leishmania* RPA1. We performed EMSA and MST experiments

**Fig. 5 | MST confirms that JC-229 specifically inhibits the ssDNA-binding activity of *Tb*RPA.** MST was performed with purified RPA proteins shown in Fig. 4i, j. The ssDNA-binding activity of recombinant RPA proteins (graphs **a**, **c**, **e**, **g**) and inhibition of the activity by JC-229 (graphs **b**, **d**, **f**, **h**) are shown. **a** *Tb*RPA1 DBD-AB proteins were twofold serially diluted starting with a 1000 nM concentration and then incubated with 10 nM oligo dT$_{32}$ labeled with 5′Cy5 (5′Cy5-dT$_{32}$). The binding was analyzed by MST. **b** Serial dilutions of JC-229 were pre-incubated with 30 nM *Tb*RPA1 DBD-AB protein and then with 10 nM 5′Cy5-dT$_{32}$. Inhibition of binding was analyzed by MST. **c** Recombinant *Tb*RPA complex was twofold serially diluted from 400 nM concentration and these were incubated with 10 nM 5′Cy5-dT$_{32}$. The binding was analyzed by MST. **d** Serial dilutions of JC-229 were pre-incubated with 25 nM *Tb*RPA complex and then with 10 nM 5′Cy5-dT$_{32}$. Inhibition of binding was analyzed by MST. **e**, **f** *Hs*RPA1 DBD-AB protein was examined as in Fig. 5a, b by MST (**e**: ssDNA-binding activity, **f**: JC-229 inhibition). **g**, **h** *Hs*RPA complex was examined as shown in Fig. 5c, d. (**g**: ssDNA-binding activity, **h**: JC-229 inhibition). Three independent MST experiments were performed for Fig. 5a–h (*n* = 3). Error bars indicate mean ± SD. Statistical analysis and plotting were performed with GraphPad Prism software. $K_d$ and IC$_{50}$ values were obtained with one site binding total and standard 4-PL curve, respectively. Source data are provided as a Source Data file.

using recombinant *Tc*RPA1 DBD-AB and *L. mexicana* RPA1 DBD-AB proteins tagged with 6xHis at the N-terminus (Fig. 6i and Supplementary Fig. 16a, b). The DBD-AB of both *Tc*RPA1 and *Lmex*RPA1 showed strong ssDNA-binding activities in EMSA (Fig. 6a, c) and MST (with $K_d$ values of 12.6 nM for *Tc*DBD-AB, 13.2 nM for *Lmex*DBD-AB, Fig. 6e, g). However, to our surprise, JC-229 did not inhibit either RPA1s both in EMSA (Fig. 6b, d) and MST (Fig. 6f, h). ESMA data were quantified and plotted in Supplementary Fig. 16c–f.

To explain the differential effects of JC-229 on these highly conserved trypanosome orthologs, we compared the potential binding sites of JC-229 in the DBD-A of *Tb*, *Tc*, and *Lmex*RPA1 (Fig. 7a, b). Homology models of *Tc*DBD-A and *Lmex*DBD-A were generated and aligned with JC-229 bound *Tb*DBD-A structure model. We predicted that the amino acid at position 105 is critical for differential effects: Serine 105 (S105) of *Tb*RPA1 vs. Threonine 105 (T105) of *Tc*RPA1 and *Lmex*RPA1 (Fig. 7b). No steric repulsion between JC-229 and the surface of S105 in *Tb*DBD-A is expected based on the electrostatic potential surface of S105 in the docking model (Fig. 7b(i)). However, the methyl group of T105 of *Tc* or *Lmex*DBD-A could cause van der Waals repulsion with JC-229 in the aligned structures (Fig. 7b(ii), (iii)). To validate our rationale for no inhibition to *Tc* or *Lmex*RPA1, we purified *Tb*RPA1 DBD-AB-S105T and *Tc*RPA1 DBD-AB-T105S mutant proteins (Fig. 7c and Supplementary Fig. 17a, b) and analyzed them using EMSA and MST (Fig. 7d–k and Supplementary Fig. 17c–f). While *T. brucei* S105T mutant protein showed a WT level of ssDNA-binding activity with a $K_d$ of 21.9 nM (vs. 24.6 nM in WT), JC-229 did not inhibit its ssDNA-binding activity (Fig. 7f, g). This result agrees well with our docking models (Fig. 7b), confirming that the binding pocket of JC-229 is in the DBD-A of *Tb*RPA1 around S105, although its hydroxyl group is not directly involved in the interaction with JC-229. The ssDNA-binding activity of *Tc*RPA1 DBD-AB T105S mutant protein was slightly decreased compared to WT ($K_d$ of 52.4 nM for mutant vs. 12.6 nM for WT), and it was inhibited by JC-229 at high concentrations (Fig. 7j, k). However, the inhibition against *T. cruzi* T105S mutant protein was not as robust as the effect of JC-229 on wild-type *Tb*RPA1 DBD-AB protein.

*Tb*RPA complex contains other DBDs with OB-fold structure, DBD-C in *Tb*RPA1, DBD-D in *Tb*RPA2, and DBD-E in *Tb*RPA3. They may also engage in ssDNA binding, like human RPA trimerization core[60] and JC-229 could also bind and inhibit these domains. If this is true, *Tb*RPA S105T mutant protein should be inhibited by JC-229. To see whether DBD-A is the major target among all DBDs in the *Tb*RPA complex, we purified the *Tb*RPA complex containing S105T mutation and performed EMSA and MST experiments (Supplementary Fig. 18). Binding of the mutant complex with ssDNA was comparable to that of WT *Tb*RPA complex. However, the mutant complex's ssDNA binding was not affected by JC-229 (Supplementary Fig. 18b–e in EMSA, 18 f, g in MST). These data, together with *Tb*DBD-AB S105T mutant, confirm that DBD-A of *Tb*RPA1 is the major target domain of JC-229.

To confirm the specificity of JC-229 in trypanosome cells, we generated a *T. brucei* strain expressing the S105T mutant RPA1 protein (three independent clones). *Tb*RPA1-S105T mutant grew slightly slower than WT and showed no defect in the cell-cycle progression (Fig. 8a, c). The mutation did not affect the level of *Tb*RPA1 protein nor the levels of γH2A or silent VSG3 proteins (Fig. 8b). As we expected, the S105T mutation conferred resistance to JC-229 (Fig. 8d and Supplementary Table 5), validating the in vitro findings and proving the potential of JC-229 as a chemical probe to target the function of *Tb*RPA1.

## Discussion

The functions of N-terminal DBD-F appear to have evolved from ssDNA binding to protein binding, as all DBDs, including the DBD-F, have the basic OB-fold structure, a known motif found in many DNA-binding proteins[26]. Our computational models also showed that the binding pose of ATRIP-*Hs*DBD-F resembles that of ssDNA-*Tb*DBD-A. Indeed, Frank et al. showed that an ATRIP peptide binds the human DBD-F and DBD-AB proteins with similar affinities[36], supporting our hypothesis that DBD-F inhibitors may be able to bind the other DBDs as well. Thus, existing human RPA1 inhibitors or their analogs may also bind *T. brucei* RPA1. Once identifying those that inhibit *Tb*RPA1 function, they can be tailored for *T. brucei* RPA1 protein. As a proof-of-principle, we examined the effect of human RPA1 inhibitors on *Tb*RPA in vitro and in cellulo and identified a chemical probe JC-229.

Our model structure of *Tb*RPA1 bound to JC-229 agrees well with the experimental results and could be used as a basis for further inhibitor design. In particular, the *Tb*RPA1-S105T mutant retains its affinity for ssDNA, but as proposed by the structure, the mutation eliminates JC-229-mediated inhibition of ssDNA interaction. Since *Tc*RPA1 and *Lmex*RPA1 possess T105 at the equivalent position and JC-229 does not inhibit these proteins, the methyl unit of T105 seems to completely block JC-229 binding to the ssDNA-binding pocket of these RPA1 proteins. The electrostatic potential surface of T105 in the model structures overlaps with JC-229 in the ssDNA-binding pocket. This steric hindrance from T105 may prevent JC-229 interaction. Interestingly, the *Tc*RPA1-T105S mutant is not as sensitive as WT *Tb*RPA1 to JC-229 in in vitro. In fact, inhibition of the *Tc*RPA1-T105S mutant is minimal, even at the highest concentration of JC-229, compared to the WT *Tb*RPA1, which contains S105. The IC$_{50}$ for T105S *Tc*DBD-AB was $159 \times 10^3$ nM vs. 228 nM for WT *Tb*DBD-AB. Unlike *T. cruzi*[42], *Tb*DBD-A or DBD-B alone does not bind ssDNA. Thus, these highly conserved trypanosome orthologs have, in fact, quite different binding properties, some of which we were able to spot using JC-229. Although the structural alignment of all models indicates that all the identical residues except S/T105 are oriented to the ssDNA-binding pockets of *Tb*, *Tc*, and *Lmex*RPA1 DBD-A, there might be some variations between the computer-generated models and the actual structures, and residues other than T105 may affect the ssDNA binding of *Tc*RPA1 DBD-A. We cannot completely exclude the possible effects of DBD-B on the conformation of DBD-A in these proteins. Structure studies by X-ray crystallography or cryo-EM would provide a better understanding of the molecular interactions between RPA1 protein and JC-229.

In humans and yeasts, ATM/ATR (TEL1/MEC1 in yeasts) phosphorylates many DDR proteins, including RPA1 and RPA2, on their SQ/TQ motifs (serine or threonine residue followed by glutamine)[14]. RPA1-T180 in humans (S178 in budding yeast) is phosphorylated by ATR/ATM (MEC1/TEL1 in budding yeast)[61,62]. The T180 residue (or S178) is located at the border of the DBD-F and DBD-A. RPA1 and RPA2 are phosphorylated during the S phase, and levels of phosphorylation increase further in response to DNA damage and replication stress in

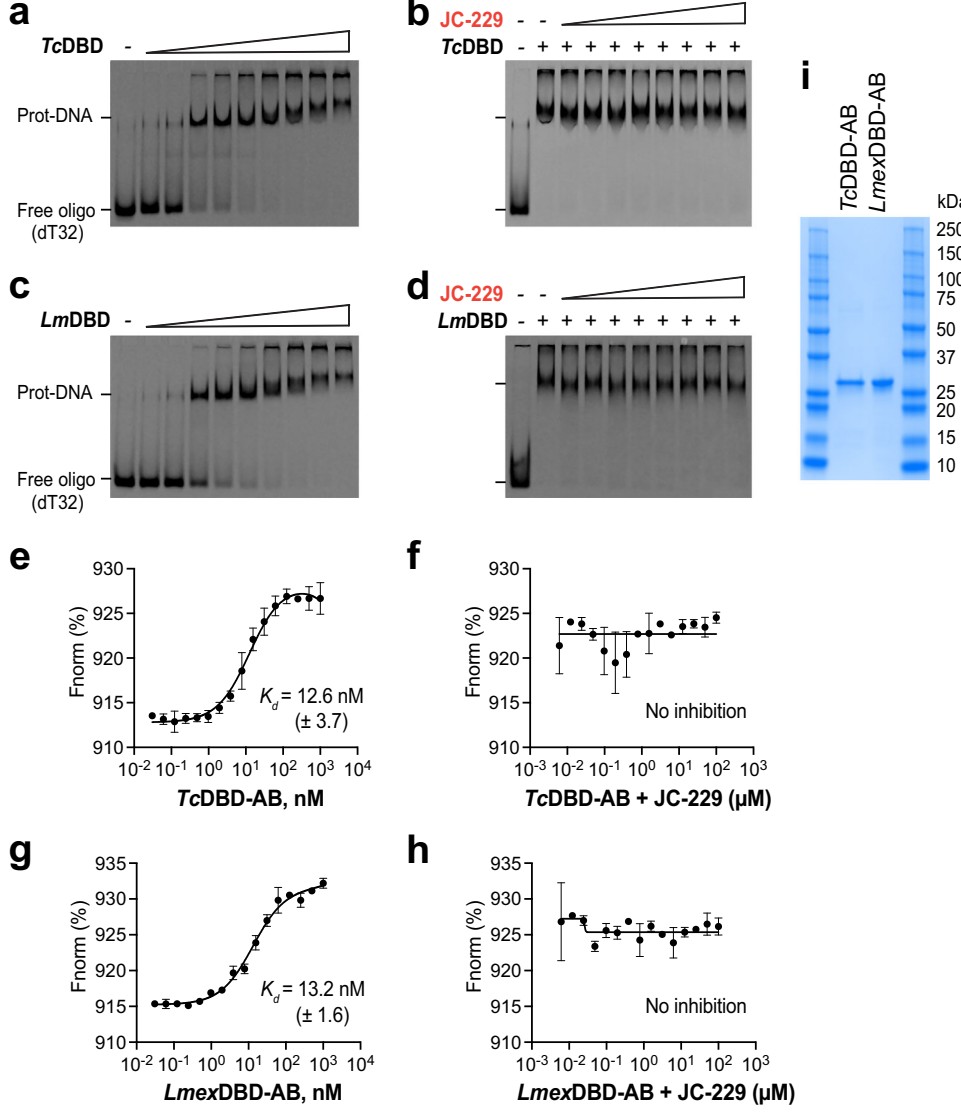

**Fig. 6 | JC-229 does not inhibit the activity of *T. cruzi* and *L. mexicana* RPA1 DBD-AB protein in vitro. a, b** EMSA was performed on recombinant *Tc*RPA1 DBD-AB as described in Fig. 4. **a** ssDNA-binding activity of *Tc*RPA1 DBD-AB. **b** Inhibition of ssDNA-binding activity of *Tc*RPA1 DBD-AB by JC-229. **c, d** EMSA assay on recombinant *Lmex*RPA1 DBD-AB. **c** ssDNA-binding activity of *Lmex*RPA1 DBD-AB. **d** Inhibition of ssDNA-binding activity of *Lmex*RPA1 DBD-AB by JC-229. Three independent EMSA experiments for Fig. 6a–d were performed with similar results. **e, f** The MST assay was performed on recombinant *Tc*RPA1 DBD-AB as described in Fig. 5. **e** ssDNA-binding activity of *Tc*RPA1 DBD-AB. **f** Inhibition of ssDNA-binding activity of *Tc*RPA1 DBD-AB by JC-229 (30 nM protein used). **g, h** MST assay performed on recombinant *Lmex*RPA1 DBD-AB. **g** ssDNA-binding activity of *Lmex*RPA1 DBD-AB. **h** Inhibition of ssDNA-binding activity of *Lmex*RPA1 DBD-AB by JC-229. Three independent MST experiments were performed for Fig. 6e–h ($n = 3$). Error bars indicate mean ± SD. Statistical analysis and plotting of MST data were performed with GraphPad Prism software. $K_d$ and $IC_{50}$ values were obtained with one site binding total and standard 4-PL curve, respectively. **i** An SDS-PAGE gel showing purified fractions used for EMSA and MST assays. Three independent experiments were performed with similar results. Source data are provided as a Source Data file.

humans and yeasts[14,63]. Using cryo-EM and FRET (fluorescence resonance energy transfer) studies in the budding yeast, Yates et al. have shown that RPA1-S178 phosphorylation is critical for the cooperative assembly of RPA complex on ssDNA[57]. Unmodified RPA binds different lengths of ssDNA probes with similar affinities, while the S178D mutant (phosphomimetic substitution) binds the longer ssDNA better. The S178D mutant RPA binds an oligo $dT_{100}$ (approximately 3 to 4 RPA molecules can bind) with a lower affinity than unmodified RPA and RPA-ssDNA complexes are more flexible with the S178D mutant. These observations indicate that phosphorylation affects the ssDNA-binding properties of RPA. In *T. brucei* RPA1, there are five SQ/TQ motifs, two are in the DBD-A (S5, S43) and three in DBD-C (T306, T341, and S355) (Supplementary Fig 1a, pink boxes). S43, T341 and S355 are conserved in *T. brucei*, *T. cruzi*, and *Leishmania*. Because our in vitro assays were performed with recombinant proteins purified from *E. coli*, all RPA

proteins should be unmodified. Whether *Tb*RPA1 phosphorylation is important for its function remains to be elucidated. Future genetic studies will provide a fuller understanding of the biological impacts of JC-229 on *T. brucei* RPA.

A foreseen issue is resistance to JC-229 (or its future analogs). The *Tb*RPA1-S105T mutant binds ssDNA with the same affinity as the WT, but it is not sensitive to JC-229 in vitro and in trypanosome cells. Because JC-229 inhibits the function of RPA by occupying the ssDNA-binding pocket, the effect of JC-229 is dependent on the level of RPA1. Therefore, mutations in the DBD-A binding pocket, mutations inducing a conformational change in the DBD-A, or mutations that increase the level of RPA could produce JC-229-resistant trypanosome cells. JC-229 seems to work rapidly, completely blocking DNA synthesis within 8 h (the earliest time point we measured) of treatment. This rapidness could reduce the chances of the *T. brucei* population

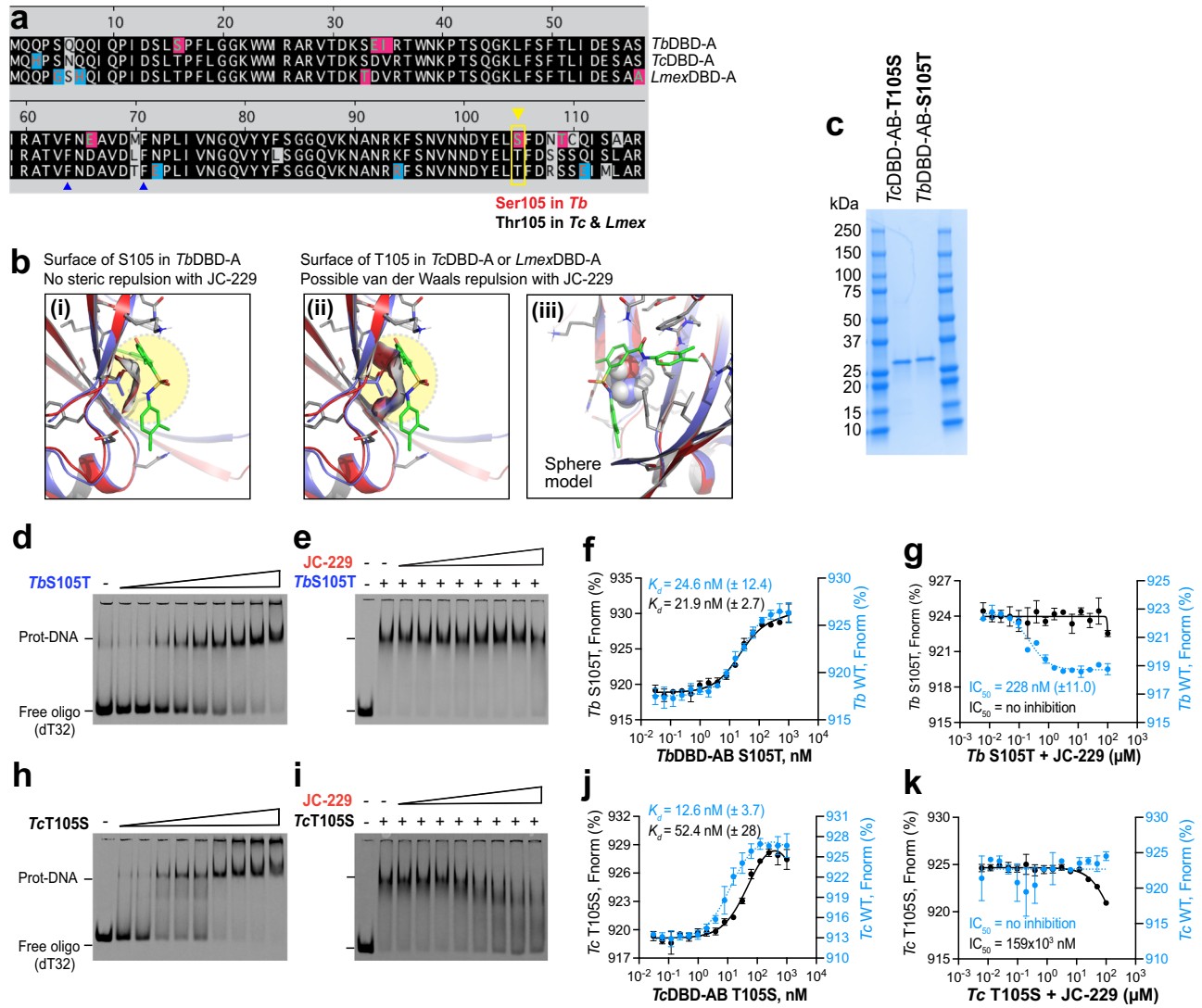

**Fig. 7 | DBD-A of *T. brucei* RPA1 has the JC-229 binding pocket and S105 is important. a** Sequence comparison of DBD-A domains of *Tb*, *Tc*, and *Lmex* RPA1. S105 of *Tb*RPA1 DBD-A and T105 of *Tc* and *Lmex*RPA1 DBD-A are shown. **b** Binding position of JC-229 with the surface of S105 in DBD-A of *Tb*RPA1 is shown in (i). The binding position of JC-229 with T105 in DBD-A of *Tc* and *Lmex*RPA1 are shown (ii: ribbon and iii: sphere model). **c** An SDS-PAGE gel showing purified fractions of *Tb*RPA1 DBD-AB S105T and *Tc*RPA1 DBD-AB T105S mutant proteins (a representative of three gels). **d**, **e** EMSA was performed as described in Fig. 4. **d** ssDNA-binding activity of *Tb*RPA1 DBD-AB S105T mutant. **e** Inhibition of ssDNA-binding activity of *Tb* S105T mutant by JC-229. **f**, **g** The MST assay was performed on recombinant *Tb*RPA1 DBD-AB S105T as described in Fig. 5. **f** ssDNA-binding activity of *Tb*RPA1 DBD-AB S105T mutant. **g** Inhibition of ssDNA binding activity of *Tb* S105T mutant

by JC-229 (30 nM protein used). **h**, **i** EMSA assay. **h** ssDNA-binding activity of *Tc*RPA1 DBD-AB T105S mutant. **i** Inhibition of the ssDNA-binding activity of *Tc* T105S mutant by JC-229. Three independent EMSA experiments were performed for Fig. 7d, e, h, i with similar results. **j**, **k** MST assay. **j** ssDNA-binding activity of *Tc*RPA1 DBD-AB T105S mutant. **k** Inhibition of ssDNA-binding activity of *Tc* T105S mutant by JC-229 (30 nM protein used). MST data of WT *Tb*RPA1 DBD-AB from Fig. 5 and WT *Tc*RPA1 DBD-AB from Fig. 6 are overlaid for comparison (blue dotted lines and circles in Fig. 7f, g, j, k). Three independent MST experiments were performed for Fig. 7f, g, j, k (n = 3). Error bars indicate mean ± SD. Statistical analysis and plotting of MST data were performed with GraphPad Prism software. $K_d$ and IC$_{50}$ values were obtained with one site binding total and standard 4-PL curve, respectively. Source data are provided as a Source Data file.

accumulating JC-229 resistant mutants, as the population doubling time of the WT bloodstream from *T. brucei* is 6 to 7 h and the accumulation of mutations depends on generation time[64]. Alsford et al. discovered potential pathways targeted by traditional anti-trypanosome drugs using a *T. brucei* RNAi library in drug resistance screens[65]. By screening genomic libraries (RNAi and overexpression libraries[50,66]) in JC-229 treated *T. brucei* cells, we could learn how JC-229 is delivered to the nucleus and what other proteins can interact with RPA genetically and/or biochemically.

JC-229 is highly effective for the inhibition of *Tb*RPA1 functions, but it needs further optimization. Although JC-229 has an IC$_{50}$ value of 24.5 nM on the *Tb*RPA complex, the EC$_{50}$ value is much higher for

*T. brucei* growth inhibition, 6.5 μM (about 265 times higher). Generally, EC$_{50}$ is higher than IC$_{50}$ due to multiple cellular factors. It is possible that only low levels of JC-229 reach the nucleus due to inefficient drug uptake. Generating more JC-229 derivatives for structure-activity relationship (SAR) analysis may yield therapeutic candidates with improved EC$_{50}$ values.

## Methods

### *Trypanosoma brucei* and human cell lines

*Trypanosoma brucei* cells were cultured in HMI-9 medium supplemented with 10% fetal bovine serum (Hyclone) at 37 °C with 5% $CO_2$, as described previously[67]. Necessary antibiotics (InvivoGen) were

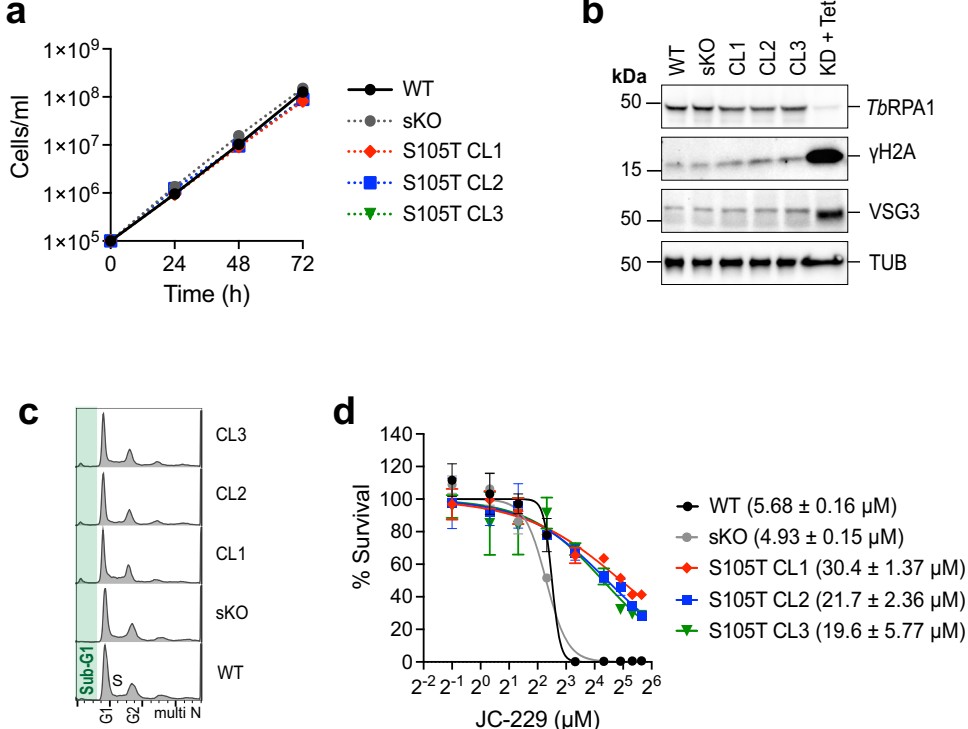

**Fig. 8 | *T. brucei* cells expressing *Tb*RPA1-S105T mutant are resistant to JC-229.** **a** Growth of WT, single knock-out (sKO), and S105T mutant trypanosome cells. Cell growth was monitored by counting cells every 24 for 72 h. Three technical replicates were used for each measurement ($n = 3$). Error bars indicate mean ± SD. **b** Immunoblot controls presenting the changes in protein levels for *Tb*RPA1, γH2A, VSG3, and Tubulin. Tubulin serves as a loading control. *Tb*RPA1-depleted cells (48 h after Tet addition) serves as a control. Three independent experiments were performed with similar results. **c** Cell-cycle profiles of *Tb*RPA1-S105T mutants. Fixed cells were stained with PI and analyzed by flow cytometry (ungated). **d** Cytotoxicity of JC-229 on *T. brucei* RPA1 S105T mutant cells, sKO and WT cells. A dose-response curve was obtained from cells treated with increasing concentrations of JC-229 (0.5, 1.25, 2.5, 5, 10, 20, 30, 40, and 50 μM) for 72 h. Three biological replicates were used for each measurement ($n = 3$) and $EC_{50}$ values were obtained from GraphPad Prism software (inhibitor vs. normalized response with variable slopes). Error bars indicate mean ± SD. Source data are provided as a Source Data file.

added at the following concentrations: G418 at 2.5 μg/ml, puromycin at 0.1 μg/ml, phleomycin at 1 μg/ml, blasticidin and hygromycin at 5 μg/ml. Cell lines used in this study are listed in Supplementary Table 6.

*Tb*RPA1 (*Tb*927.11.9130) RNAi KD cell lines were generated in the 2T1 background that expresses a T7 RNA polymerase and a Tet repressor (TetR)[50]. We first replaced the PHLEO marker in the 2T1 with NEO, generating the AMT2 strain (WT). AMT2 cells were transfected with a double-stranded RNA inducible vector (pOH10 with p2T7TA backbone) targeting the *Tb*RPA1 transcript. Two clones were obtained, AMT5 and AMT6 (denoted as KD1 and KD2).

*Tb*RPA1 single knock-out (sKO, AMT30) and S105T mutant cell lines (AMT38, 39, and 41) were generated in the SM background expressing a T7 RNA polymerase, TetR and Cre-recombinase[68,69]. To generate sKO strain, one allele of *Tb*RPA1 was deleted in the HSTB-904 strain (WT) by transfecting a knockout vector (pSUN18)−HYG-TK selection marker (hygromycin-resistance gene, HYG, conjugated with Herpes simplex virus thymidine kinase, HSV-TK) flanked by upstream and downstream homology sequences of *Tb*RPA1 gene. *Tb*RPA1-S105T knock-in (KI) vector (pAD10) was transfected to the AMT30 strain to replace the remaining WT allele of *Tb*RPA1. WT, sKO, and S105T mutant strains were PCR genotyped (Supplementary Fig. 19) and S105T mutation was confirmed by Sanger sequencing (Genewiz/Azenta).

Human cell lines, HeLa (ATCC, CRM-CCL-2) and HEK293 (ATCC, CRL-3216) were maintained in Dulbecco's modified MEM (D-MEM, GIBCO) supplemented with 10% fetal bovine serum (GIBCO) at 37 °C with 5% $CO_2$.

## Construction of plasmids for expression of recombinant protein(s)

Plasmids for the expression of *Tb*, *Tc*, *Hs*, *Lmex* RPA1 DBD-AB from *E. coli*: pSR5 (*T. brucei* RPA1 DBD-AB), pSR13 (DBD-AB of *T. cruzi* RPA1, TCDM_07088), pSR15 (*Hs*RPA1 DBD-AB), and pSR16 (DBD-AB of *L. mexicana* RPA1, LmxM.28.1820) were generated by PCR-amplifying the DBD-AB domain and cloning into a pET28a vector.

Plasmids for the simultaneous expression of three subunits of *Tb*RPA or *Hs*RPA complex in *E. coli*: To purify RPA complex from *E. coli*, we generated a vector that expresses all three subunits of the *Tb*RPA or *Hs*RPA complex, with only one subunit, RPA1, tagged with 6xHis and Thrombin cleavage site; pSR8 for *Tb*RPA and pSR18i for *Hs*RPA complex. Transcription of all three subunits is driven individually by a T7 promoter. To insert all subunits in the pET28a vector, we first amplified *Tb*RPA3 CDS (*Tb*927.9.11940) with a forward oligo containing HindIII-T7-LacO-RBS sequences and a reverse oligo containing an XhoI site. A HindIII/XhoI-digested fragment was ligated with a HindIII/XhoI-digested pET28a vector fragment, generating plasmid pSR6. Similarly, we PCR-amplified the *Tb*RPA2 CDS (*Tb*927.5.1700) with a forward oligo containing BamHI-T7-LacO-RBS sequences and a reverse oligo containing a HindIII site, then digested with BamHI and HindIII, ligated with a BamHI/HindIII-digested pSR6 vector fragment, generating plasmid pSR7 (RPA2 & RPA3 in pET28a). We then inserted the *Tb*RPA1 CDS in pSR7 using NheI/BamHI digestion and ligation, generating the final plasmid pSR8 (6xHis-*Tb*RPA1, *Tb*RPA2, and *Tb*RPA3 in pET28a).

To generate a vector for the simultaneous expression of human RPA subunits (6xHis-RPA1, RPA2, and RPA3), we used site-directed mutagenesis using two pairs of oligos and a template,

p11d-tRPA(123)[58] (AddGene). The upstream region (392 nt) of *Hs*RPA1 was amplified with a forward oligo (HK-1256) and a reverse oligo tailed with 6xHis-Thrombin cleavage site sequences (HK-1232) and a 982 bp region of *Hs*RPA1 CDS with a forward oligo tailed with 6xHis-Thrombin cleavage site (HK-1233) and a reverse oligo (HK-1257). Two PCR products were pooled at equimolar ratio and amplified with HK-1256 and HK-1257 oligos, which resulted in the insertion of 6xHis-Thrombin cleavage sequences at the N-terminus of *Hs*RPA1 (pSR18i).

Plasmids for the expression of DBD-AB mutant proteins from *E. coli*, *Tb*RPA1 DBD-AB with F64A or W188A or S105T and *Tc*RPA1 DBD-AB with T105S: Site-directed mutagenesis (Quick change kit, NEB) was performed to insert an indicated mutation to *Tb*DBD-AB in plasmid pSR5 (to introduce F64A, W188A, or S105T mutation) or to *Tc*DBD-AB in plasmid pSR13 (to introduce T105S mutation).

Plasmids for the expression of *Tb*RPA1 DBD-A and DBD-B from *E. coli*: pSR28 (*T. brucei* RPA1 DBD-A) and pSR29 (*T. brucei* RPA1 DBD-B) were generated by PCR-amplifying the DBD-A or DBD-B domain and cloning into a pET28a vector.

Plasmids for the expression of *Hs*RPA1 DBD-F from *E. coli*: pSR31 (*Hs*RPA1 DBD-F) was generated by PCR-amplifying the DBD-F domain and cloning it into a pET28a vector.

A plasmid for the expression of the *Tb*RPA complex containing the S105T mutation: pRS8 was used to incorporate the S105T mutation in the *Tb*RPA1 gene.

Plasmids are listed in Supplementary Table 7 and oligonucleotides in Supplementary Table 8.

### Cell growth and viability assay

To determine the cell growth rate of *Tb*RPA1-depleted cells, WT and two KD strains (KD1 and KD2) were seeded at a density of $1 \times 10^5$ cells/ml and cultured with or without tetracycline for the induction of RNAi. Cells were counted every 12 h for 2 days. To determine the viability of *Tb*RPA1-depleted cells, we carried out the AlamarBlue assay (Thermo Fisher, catalog number A50101), an established colorimetric technique that detects metabolically active cells by measuring resorufin converted from resazurin by reducing the environment of live cells[70]. Cells from each time point were incubated with AlamarBlue dye for 4–6 h, and the percentage of live cells was determined[70]. Synergy H1 microplate reader (BioTek) was used with 560/590 nm (excitation/emission) filter settings and relative fluorescence unit (RFU) values were obtained. The percentage of viable cells were calculated from the RFU values.

To determine the growth rate of WT, sKO, and three *Tb*RPA1-S105T mutant strains, cells were seeded at a density of $1 \times 10^5$ cells/ml, counted every day for 3 days and examined as above.

### Cell-cycle analysis and BrdU pulse experiments using flow cytometry[49]

About 10 million cells were collected, fixed with ice-cold 70% ethanol, and incubated with 50 µg/ml PI (Sigma-Aldrich) and 200 µg/ml RNase A (Sigma-Aldrich) in 1x PBS (Corning) at 37 °C for 30 min in the dark. Samples were analyzed by flow cytometry (FACSVia, BD Biosciences) and FlowJo.

The 5-Bromo-2′-deoxyuridine (BrdU) incorporation assay was performed as described previously in ref. 49. BrdU (Sigma-Aldrich, catalog number B5002) was added to cell cultures to a final concentration of 500 µM and incubated for 40 min at 37 °C. About 20 million cells were harvested and fixed in ice-cold 70% ethanol. DNA in fixed cells was denatured in 0.1 N HCl (Fisher Scientific) with 0.5% Tween 20 (Sigma-Aldrich) and 1% BSA (Jackson Immunoresearch) for 20 min at room temperature. To stain newly incorporated DNA, fixed cells were incubated with 1 µg/ml mouse anti-BrdU antibody (BD Pharmingen, catalog number 555627, lot 6084615, dilution 1:500) at room temperature for 2.5 h and then with 4 µg/ml donkey anti-mouse

Alexa 488 (Invitrogen, catalog number A-21202, lot 94C2-1, dilution 1:500) at room temperature for 1 h in the dark. Bulk DNA was stained with 5 µg/ml PI by incubating at room temperature for 30 min in the dark. Samples were analyzed by flow cytometry (FACSVia, BD Biosciences) and FlowJo.

### Microscopy

For microscopic analysis, untreated or JC-229 treated cells were seeded, and fixed with 1% paraformaldehyde for 10 min at room temperature. Cells were then washed with 1x PBS and stained with DAPI (0.5 µg/ml) (Sigma). Images were captured using fluorescence microscopy (EVOS M5000, Thermo Fisher Scientific) at 40x magnification and edited with Adobe Photoshop.

### Immunoblotting

Cells were harvested at indicated time points and suspended in Laemmli buffer (Bio-Rad) at 0.25 million cells per µl. Denatured proteins from whole cells were separated on a 4–20% gradient SDS-PAGE gel (Bio-Rad) and transferred to a nitrocellulose membrane (GE Healthcare). The membranes were probed with polyclonal antibodies against *Tb*RPA1 (a gift from Bibo Li laboratory, ProSci Inc., ID: 21371, dilution factor 1:1,000) or γH2A (generated by ABclonal for this study, antibody ID E16070 (P), dilution factor 1:1,000), or monoclonal mouse antibodies against VSG3 (Antibody and Bioresource Core facility, MSKCC, catalog number cloneVSG224-11D6, lot EQ.11-Pur.-A, dilution factor 1:1,000) or Tubulin (Sigma, catalog number T7451, clone 6-11B-1, lot 0000149704, dilution 1:5000) in 1x PBS with 0.1% Tween 20 and 1% milk (Bio-Rad) at 4 °C overnight. The membranes were then incubated with secondary antibodies conjugated with Horseradish Peroxidase (HRP) (GE Healthcare; Rabbit HRP: catalog number NA934V, lot 17640116; Mouse HRP: catalog number NA931V, lot 17041904, dilution 1:5000) for the detection of corresponding proteins using chemiluminescence (Hyglo ELC solution from Denville Scientific, catalog number E2400; ChemiDoc Imaging System from Bio-Rad).

### Cell viability of JC-229-treated *T. brucei* and human cells

The initial assessment of JC-229's inhibitory effect on *T. brucei* cell growth was performed at 50 µM concentration. Trypanosome cells were seeded at $1 \times 10^3$ cells/ml density in 180 µl in a 96-well culture plate in triplicate. These cells were incubated with 50 µM JC-229 for 24, 48, and 72 h. DMSO was used as a vehicle control at 0.0075%. JC-229-treated cultures and control cultures were analyzed with the AlamarBlue dye as described above.

To obtain the $EC_{50}$ value of JC-229, *T. brucei* cells were seeded at $1 \times 10^3$ cells/ml density in 180 µl in a 96-well culture plate in triplicate. These cells were incubated for 72 h with increasing concentrations of JC-229 (0.5 to 50 µM) and cell viability was determined using the AlamarBlue dye as described above.

To determine the inhibitory effect of JC-229 on human cells, HeLa and HEK293 cells were suspended in 180 µl D-MEM with 10% FBS and plated at a density of $2 \times 10^4$ cells/ml in triplicate. Cells were adhered overnight, the medium was replaced, and cells were then incubated for 72 h with increasing concentrations of JC-229 (0.5 to 50 µM), and cell viability was determined using the AlamarBlue dye as described above. Percent viability was plotted using GraphPad Prism software. Curves generated from human cell data did not fit the 4-PL, as they did not reach the bottom plateau with the highest concentration of JC-229 used. We estimated the $EC_{50}$ values using the concentration of JC-229 that causes a 50% reduction of cell viability (inhibitor vs. normalized response, variable slope).

### Structure modeling of RPA1 proteins and computational analysis of induced-fit model

The homology model of *Tb*RPA1 was built by a protein structure homology-modeling server (SWISS-MODEL). Template search

showed that the 4gop.1.C structure (*Ustilago maydis* RPA1 and ssDNA complex) has 36% sequence identity with *Tb*RPA1, and it was selected for building the homology model: 4GOP [https://doi.org/10.1101/gad. 194787.112]. The model structure was refined with Protein Preparation Wizard implemented in Maestro 12. The protein structure was imported into a workspace and preprocessed to assign bond orders, add hydrogen atoms, create zero-order bonds to metals, create disulfide bonds, and delete water molecules beyond 5 Å from hetero groups. In addition, missing atoms in residues and missing loops were added using Prime to generate a complete protein structure. The protein structure was further refined via automated H-bond assignment and restrained minimization with OPLS 2005 force field by converging heavy atoms to 0.3 Å RMSD. DBD-A and DBD-B domains of the refined *Tb*RPA1 model structure were selected for Induced Fit docking of JC-229. Initially, the ligand structure (JC-229) was built using the 3D build module and prepared by the Ligprep module for molecular docking. Induced-Fit docking was performed using the 'standard protocol' (SP) mode, and T62, F64, and K89 were selected to define the receptor grid. The size of the inner grid box was set to 10 Å (default value), and its outer grid box was automatically determined, which includes most part of the DBD-A structure. Ligand ring conformation was sampled with a 2.5 kcal/mol energy window. Residues within 5 Å of the ligand in the initial docking were given flexibility during the prime refinement process, and the ligand was re-docked using the Glide SP mode. The induced-fit docking (IFD) score was calculated by the Prime energy, glide score, and glide Coulomb energy. The final binding pose was selected based on the IFD scores and the binding poses of Induced-Fit docking models.

The AlphaFold structure of *Tb*RPA1 was obtained from AlphaFold Protein Structure Database[71] (identifier: AF-Q384B5-F1, V4), and its DBD-A and DBD-B domains were aligned to SWISS-models using Pymol, respectively.

### Purification of recombination proteins

A protein expression plasmid was transformed into BL21 competent cells (Invitrogen) and selected in an appropriate antibiotic (ampicillin or kanamycin). Expression of a recombinant protein was induced with Isopropyl ß-D-1-thiogalactopyranoside (IPTG) at 1 mM concentration. Induction was carried out for 3 h at 37 °C for all DBD-AB, *Tb*DBD-A, *Tb*DBD-B, and *Hs*DBD-F proteins and for 16 h at 18 °C for *Tb*RPA complex and *Hs*RPA complex. About 50 ml of IPTG-induced cells were harvested by centrifugation at 16,000 × g at 4 °C and aliquoted (five microcentrifuge tubes), snap-frozen, and stored at −80 °C. For purification, frozen cell pellets were resuspended and lysed in B-PER™ Complete Bacterial Protein Extraction Reagent (Thermo Fisher Scientific, catalog number 89821) with 1 mM phenylmethylsulfonylfloride (PMSF) and EDTA-free Protease/phosphatase-inhibitor cocktail (Thermo Fisher Scientific, catalog number 1861280) for 10 min with inverting the tubes a few times. The whole cell lysate was centrifuged at 16,000 × g at 4 °C. The supernatant diluted with an equal volume of Equilibrium Buffer (2x PBS with 10 mM imidazole, pH 7.4) was mixed with Ni-NTA resin (washed twice with Equilibrium Buffer) and incubated for 2 h at 4 °C on a tube rotator. Unbound fraction (flow-through) was collected by centrifugation at 1000 × g for 1 min at 4 °C. The resin was washed four times with Wash Buffer (2x PBS with 25 mM imidazole). 6xHis-tagged proteins were eluted with Elution Buffer (2x PBS with 500 mM imidazole). Samples from each step were analyzed on a 4–20% SDS-PAGE gel. Eluted fractions containing a high amount of recombinant protein were desalted by gel-filtration chromatography (Zeba Spin Desalting Column, Thermo Fisher Scientific, catalog number 89890). Purified proteins were snap-frozen and stored at −80 °C and used for assays below.

### Nano-liquid chromatography-tandem mass spectrometry (LC-MS/MS)

Purified 6xHis-tagged *Tb*RPA protein were concentrated using a 50 kDa MWCO centrifugal concentrator (Sigma/Millipore) and ran onto SDS-PAGE (NuPAGE™ 10% Bis-Tris Gel, Invitrogen). Desired gel bands were sliced, and in-gel digestion was performed separately with sequencing grade trypsin (Thermo Fisher Scientific) and analyzed by Nano-LC-MS/MS as described previously in ref. 72. Samples were analyzed using a Q Exactive HF tandem mass spectrometer coupled to a Dionex Ultimate 3000 RLSCnano System (Thermo Fisher Scientific). To initiate analysis, samples were loaded onto a fused silica trap column Acclaim PepMap 100, 75 um × 2cm (Thermo Fisher Scientific). After washing for 5 min at 5 μl/min with 0.1% TFA, the trap column was brought in-line with an analytical column (Nanoease MZ peptide BEH C18, 130 A, 1.7 um, 75 um × 250 mm, Waters) for LC-MS/MS. Peptides were eluted using a segmented linear gradient from 4 to 90% B (A: 0.2% formic acid, B: 0.08% formic acid, 80% ACN): 4–15% B in 5 min, 15–50% B in 50 min, and 50–90% B in 15 min. Mass spectrometry data were acquired using a data-dependent acquisition (DDA) procedure with a cyclic series of a full scan from 250–2000 with a resolution of 120,000 followed by MS/MS (HCD, relative collision energy 27%) of the 20 most intense ions (charge +1 to +6) and a dynamic exclusion duration of 20 s. Major peptides were confirmed manually.

Database Search: The peak list of the LC-MS/MS were generated by Thermo Proteome Discoverer (v. 2.1) into MASCOT Generic Format (MGF) and searched against custom supplied sequences, *E. Coli* plus a database composed of common lab contaminants using in-house version of X!Tandem. Search parameters are as follows: fragment mass error: 20 ppm, parent mass error: ±7 ppm; fixed modification: none; flexible modifications: Oxidation on Methionine for the primary search and dioxidation of methionine and glutamine to pyro-glutamine at the refinement stage. Only two miss-cleavages of trypsin were allowed for the primary search and five miss-cleavages were allowed for semi-tryptic cleavage at the refinement stage. Only spectra with loge <−2 were included in the final report. Peptides belonging to target proteins were manually inspected to determine the exact processed sequences.

### Size exclusion chromatography (SEC)

Purified 6xHis-tagged *Tb*RPA protein were loaded onto a Superdex 200 10/300 column (GE Healthcare) equilibrated with 2x PBS. The column was calibrated using a gel-filtration standard (Bio-Rad) containing a mixture of thyroglobulin, bovine γ-globulin, chicken ovalbumin, equine myoglobin, and vitamin B12. Peak fractions of the eluted protein were analyzed by SDS-PAGE.

### Mass photometry (MP)

Mass photometry data was acquired on a Refeyn One^MP instrument (Refeyn Ltd) as described previously with some modification[73]. Briefly, the microscopic coverslips (Thorlabs) and gaskets (Grace bio-Labs) were cleaned with 100% IPA, followed by three times washing in ddH$_2$O and dried with HEPA-filtered air. For calibration, measurement of a mixture of known protein oligomer solutions of BSA, thyroglobulin, and beta-amylase (Sigma) were done with a final concentration of 40, 5, and 10 nM, respectively. For focusing the instrument, 12 μl of 2x PBS was added to each well followed by the addition of 8 μl *Tb*RPA protein (final concentration = 20 nM). Movies were acquired for 60 s using Refeyn Acquire^MP (v. 2022 R1; Refeyn Ltd) using standard settings by the manufacturer. All movies were processed, analyzed, and mass was estimated by fitting a Gaussian distribution to the data using Refeyn Discover^MP (v. 2022 R1; Refeyn Ltd).

### Electrophoretic mobility shift assays (EMSA)

ssDNA-binding assay: 5′IRDye800-labeled oligos (dT$_{32}$, dT$_{20}$, and dT$_{12}$) were purchased from Integrated DNA Technologies (IDT). Increasing

concentrations of recombinant protein (from 0.05 to 100 nM) were incubated with 2 nM of 5′IRDye800-labeled oligo in binding buffer (20 mM Tris, pH 7.5, 10 mM NaCl, 10 mM MgCl$_2$, 2 mM dithiothreitol (DTT), 10% glycerol, and 0.1 mg/ml BSA) for 15 min at room temperature[74]. About 4 µl of 6x Orange dye (Sigma) was added to the 20 µl reaction, and from that, 8 µl was separated on a non-denaturing polyacrylamide gel in 1x Tris-Borate-EDTA (TBE, Invitrogen) buffer; 4% gel for RPA complex and 8% for DBD-AB, DBD-A, or DBD-B proteins. Gels were pre-run for 1 h at a constant 100 V. Protein−ssDNA complex and unbound ssDNA were separated by running the gel at a constant 80 V at 4 °C in the dark and then visualized using LI-COR Odyssey imaging system and quantified with ImageJ software.

Inhibition of DNA-binding activity by JC-229: To determine the inhibitory effect of JC-229, increasing concentrations of JC-229 (from 2.5 to 320 µM) were incubated with recombinant protein (50 nM for DBD-AB and 12.5 nM RPA) in 10 µl reaction volume for 15 min at room temperature. To each of the reaction mix, 10 µl of 5′IRDye800-labeled oligo was added to a final concentration of 2 nM and the reaction was then incubated for another 15 min at room temperature. Samples were separated on a non-denaturing polyacrylamide gel and analyzed as described above in ssDNA-binding assay.

## Microscale thermophoresis (MST)

MST experiments were performed on Monolith NT.115 instrument with a red filter (NanoTemper Technologies) in binding buffer using premium-coated capillaries (NanoTemper Technologies). MST experiments were performed with 40% LED power and 40% MST power at 22 °C.

ssDNA-binding assay: The 5′Cy5-labeled oligos (dT$_{32}$, dT$_{20}$, and dT$_{12}$) were purchased from IDT. Increasing concentrations of purified protein (from 0.06 to 2000 nM for DBD-AB, DBD-A, and DBD-B proteins; from 0.02 to 800 nM for RPA complexes) were incubated with equal volume (10 µl) of 5′ Cy5-labeld oligo (10 nM final concentration) in ssDNA binding buffer (20 mM Tris, pH 7.5, 10 mM NaCl, 10 mM MgCl$_2$, 2 mM DTT, 10% glycerol, and 0.1 mg/ml BSA) for 15 min at room temperature. Samples were then loaded into MST capillaries.

Inhibition of DNA-binding activity by JC-229: Increasing concentrations of JC-229 (from 0.024 to 400 µM) were incubated within a range of 15 to 30 nM recombinant protein in 10 µl volume for 15 min at room temperature. To these, 10 µl of 5′Cy5-labeled oligo was added to a final concentration of 10 nM and incubated for another 15 min at room temperature. Samples were then loaded into MST capillaries.

HsDBD-F and ATRIP binding assay[33]: Increasing concentrations of purified HsDBD-F protein or BSA (from 9 to 30,000 nM) was incubated with equal volume (10 µl) of Cy5-labeled ATRIP peptide (50 nM final concentration) in the binding buffer (50 mM HEPES, pH 8.0, 75 mM NaCl, 5 mM DTT, pH 7.5, and 0.05% Tween 20) for 15 min at room temperature. Samples were then loaded into MST capillaries.

Inhibition of the HsDBD-F binding with ATRIP by JC-229: Increasing concentrations of JC-229 (from 0.024 to 400 µM) were incubated with 80 µM recombinant HsDBD-F in 10 µl volume for 15 min at room temperature. To these, 10 µl of Cy5-labeled ATRIP peptide was added to a final concentration of 50 nM and incubated for another 15 min at room temperature. Samples were then loaded into MST capillaries.

We used one site binding to obtain a dissociation constant ($K_d$) and standard 4-parameter logistic curve (Sigmoidal curve) analysis to obtain the IC$_{50}$ value of JC-229. We used GraphPad Prism software.

## General experimental details for the synthesis of chemical agents

Commercially available reagents were used without further purification. Information on all commercial chemicals and solvents used in the synthesis of chemical agents are provided in the source file. The term "concentrated under reduced pressure" refers to the removal of solvents and other volatile materials using a rotary evaporator with the water bath temperature below 40 °C, followed by the removal of residual solvent at high vacuum (<0.2 mbar). Proton nuclear magnetic resonance ($^1$H NMR) spectra were recorded on a Bruker 400 MHz instrument. Carbon-13 nuclear magnetic resonance ($^{13}$C NMR) spectra were recorded on a Bruker 101 MHz instrument. The proton signal for residual non-deuterated solvent (δ 2.50 for dimethyl sulfoxide) was used as an internal reference for $^1$H NMR spectra. For $^{13}$C NMR spectra, chemical shifts are reported relative to the δ 39.52 resonance of dimethyl sulfoxide. Coupling constants are reported in Hz. Analytical thin layer chromatography (TLC) was performed on silica gel 60 F$_{254}$ glass plates precoated with a 0.210 to 0.270 mm thickness of silica gel. The TLC plates were visualized with UV light. Column chromatography was performed using silica gel 60 (40–60 Angstroms) purchased from Oakwood. Shimadzu preparative HPLC and analytical UFLC systems were used for purifying and measuring the purity of the final compound, respectively.

## Synthesis of JC-229

*N-3-(chlorosulfonyl)−4-methylbenzoic acid* (intermediate 1): Chlorosulfonic acid (6 ml, 90 mmol, 13 eq) was added to 4-methylbenzoic acid (1 g, 7.34 mmol) and the mixture was heated at 100 °C for 2 h. The reaction was then cooled to room temperature and ice water was added to the mixture until white precipitates formed. The white precipitate was collected and dried under vacuum to give the product (1.44 g, 84% yield). $^1$H NMR (400 MHz, DMSO-$d_6$): δ 8.32 (d, $J$ = 1.9 Hz, 1H), 7.77 (dd, $J$ = 7.8 Hz, 2.0 Hz, 1H), 7.26 (d, $J$ = 7.9 Hz, 1H), 2.58 (s, 3H). The $^1$H NMR spectroscopy data matches with the report[75].

*3-(chlorosulfonyl)−4-methylbenzoyl chloride* (intermediate 2): Thionyl chloride (12 ml, 169.5 mmol, 77 eq) was added to 3-(chlorosulfonyl)−4-methylbenzoic acid (1) (500 mg, 2.13 mmol), and the reaction mixture was heated at 75 °C for 12 h. The thionyl chloride was distilled to produce the titled product as a red oil, which was used for the next reaction without further purification.

*N-(3,4-dichlorophenyl)−3-(N-(3,4-dichlorophenyl)sulfamoyl)−4-methylbenzamide* (JC-229): 3,4-dichloroaniline (691 mg, 4.27 mmol) was added to a solution of 3-(chlorosulfonyl)−4-methylbenzoyl chloride (180 mg, 0.71 mmol) in toluene (5 ml). The reaction mixture was heated at 75 °C for 12 h. The reaction was then cooled to room temperature and diluted with ethyl acetate and washed with saturated NaHCO$_3$, brine, and the organic layer was dried over MgSO$_4$ and concentrated under reduced pressure. The crude mixture was purified using flash chromatography using hexane and ethyl acetate and further purified using preparative HPLC, providing the desired product (JC-229) as an off-white solid (297 mg, 83% yield, >98% purity by LC). $^1$H NMR (400 MHz, DMSO-$d_6$): δ 11.04 (s, 1H), 10.69 (s, 1H), 8.51 (d, $J$ = 2.0 Hz, 1H), 8.14 (dd, $J$ = 6.9 Hz, 2.2 Hz, 2H), 7.76 (dd, $J$ = 8.9 Hz, 2.4 Hz, 1H), 7.62 (dd, $J$ = 14.2 Hz, 8.5 Hz, 2H), 7.50 (d, $J$ = 8.8 Hz, 1H), 7.28 (d, $J$ = 2.6 Hz, 1H), 7.08 (dd, $J$ = 8.8 Hz, 2.6 Hz, 1H), 2.66 (s, 3H); $^{13}$C NMR (101 MHz, DMSO-d$_6$) δ 163.97, 140.88, 138.91, 137.46, 137.31, 133.10, 132.28, 132.24, 131.51, 131.32, 130.85, 130.59, 128.88, 125.57, 125.41, 121.63, 120.41, 119.81, 118.45, 19.70; HRMS (ESI-TOF): $m/z$ [M + H]+ calculated for C20H15Cl4N2O3S, 504.9528; found, 504.9528.

## Synthesis of JC-230, JC-231, JC-232, and JC-233

JC-230 was synthesized by following the procedures in Supplementary Fig. 2a[35]. Briefly, the hydrazide intermediate (2), made from the corresponding methyl ester (1) and hydrazine, was coupled with 1,2-dichloro-4-isothiocyanatobenzene to the thiosemicarbazide (3). This intermediate was cyclized under basic conditions to the thiotriazole (4), of which alkylation with 2-bromoacetic acid produced the desired product (JC-230). In addition, JC-231, JC-232, and JC-233 were synthesized by following the procedures in Supplementary Fig. 2b[37]. Briefly,

Knorr pyrazole synthesis was applied using **6** and (3,4-dichlorophenyl) hydrazine (HCl salt form) to prepare the 1,5-biarylpyrazole intermediates (**7**), which were hydrolyzed to JC-231 and JC-233. JC-232 was made from the reduction of **7a** with Raney nickel followed by hydrolysis. The structures of these agents were confirmed by $^1$H and $^{13}$C NMR and mass spectrometry analysis.

## Solid phase synthesis of ATRIP-ε-Acp-Cy5 probe

The peptide was synthesized in a disposable 24 ml polypropylene syringe fitted with a PTFE filter using 500 mg (1 eq.) TentaGel S RAM resin (0.22 mmol/g). The resin was swelled in DMF for 2 h at 25 °C, followed by fluorenylmethoxycarbonyl (Fmoc) group deprotection by 20% piperidine in dimethylformamide (DMF) (2 × 30 min incubation, 10 ml each). After deprotection, the resin was washed with DMF (6 × 6 ml), dichloromethane (DCM) (6 × 6 ml), and DMF (6 × 6 ml) [coupling and Fmoc-deprotection reactions]. To an empty glass vial were added Fmoc-protected amino acid (Fmoc-L-Leucine, 5 eq.), 1-hydroxybenzotriazole hydrate (HOBt, 5 eq.), and 1-[Bis(dimethylamino)methylene]−1H-1,2,3-triazolo[4,5-b]pyridinium 3-oxide hexafluorophosphate (HATU, 5 eq.) in 10% *N,N*-diisopropylethylamine (DIPEA)/90% DMF (10 ml) for pre-activation for 15 min. The preactivated solution was added to the syringe containing the resin, which was shaken for 2.5 h at 25 °C. The solution was drained from the syringe, and the resin was washed with DMF (6 × 6 ml), DCM (6 × 6 ml), and DMF (6 × 6 ml). After the coupling reaction with the first amino acid, the resin was treated with Ac₂O/DIPEA/DMF (3/2/5, 10 ml) for 30 min in order to cap any remaining free amine. After the Fmoc-protected amino acid was built on the resin, Fmoc-deprotection was carried out using 20% piperidine in DMF (2 × 30 min, 10 ml) at 25 °C. The coupling reaction and Fmoc-deprotection were monitored by Ninhydrin Kaiser tests, analytical LC by cleavage of around 10–15 beads of the resin, and mass spectrometry analysis. The coupling reaction and Fmoc-deprotection were repeated with the following Fmoc-protected amino acids in sequence: Fmoc-Ala-OH, Fmoc-Phe-OH, Fmoc-Trp(Boc)-OH, Fmoc-Glu(OtBu)-OH, Fmoc-Glu(OtBu)-OH, Fmoc-Leu-OH, Fmoc-Asp(OtBu)-OH, Fmoc-Asp(OtBu)-OH, Fmoc-L-OH, Fmoc-Thr(OtBu)-OH, Fmoc-Phe-OH, and Fmoc-Asp(OtBu)-OH. Once the Fmoc-protected peptides were built on the resin, 20% piperidine in DMF (2 × 30 min, 10 ml) was added for Fmoc-deprotection. After draining the solution and washing the resin with DMF (6 × 6 ml), DCM (6 × 6 ml) and DMF (6 × 6 ml), the Fmoc-ε-Acp-OH (5 eq.), HOBt (5 eq.), and HATU (5 eq.) in DIPEA/DMF (1/9, 10 ml) was pre-activated for 15 min followed by coupling with the ATRIP peptide on the resin for 2.5 h at 25 °C, of which Fmoc was deprotected with 20% piperidine in DMF (2 × 30 min, 10 ml each). The deprotected N-terminal ATRIP-ε-Acp was coupled with the Cy5-NHS ester (1.2 eq) in anhydrous DMF containing 20% DIPEA in the dark for 6 h at 25 °C. The solution was drained, and the resin was washed with DMF (6 × 6 ml), DCM (6 × 6 ml), and DMF (6 × 6 ml). The resin was treated with a cleavage solution of trifluoroacetic acid (TFA):triisopropylsilane (TIPS):H₂O (95:2.5:2.5) for 2 h at 25 °C. The reaction solution containing ATRIP-ε-Acp-Cy5 probes was drained into the cold ether, and the precipitates were formed and collected by centrifugation. The precipitates were dissolved by MeOH and purified by preparative HPLC with a dual pump Shimadzu LC-20AP system. The HPLC system was equipped with a SunFire C18 preparative column (19 × 250 mm, 10 microns) at λ = 220 nm, with a mobile phase of (A) H₂O (0.1% TFA)/(B) methanol (MeOH)/acetonitrile (ACN) (3:1) (0.1% TFA), at a flow rate of 60 ml/min with 10%(B) for 30 s, a gradient up to 90%(B) for 9.5 min, and 90%(B) for 3 min. A fraction containing the desired product was collected and lyophilized to afford the ATRIP-ε-Acp-Cy5 probe (H₂NCO-LAFWEELDDATFD-ε-Acp-Cy5) as a blue solid, which was characterized by high-resolution Mass Spectrometer (HRMS) and analytical Liquid Chromatography (LC) at λ = 650 nm: HRMS (ESI-TOF): *m/z* [M]⁺ calculated for $C_{111}H_{147}N_{18}O_{26}^+$, 2148.0728;

found, 2148.0699. The fraction also contained a by-product (H₂NCO-LAFWEELDDAATFD-ε-Acp-Cy5) detected by the HRMS analysis: HRMS (ESI-TOF) *m/z* [M]⁺ calculated for $C_{114}H_{152}N_{19}O_{27}^+$, 2219.1100; found 2219.1077 (Supplementary Fig. 15).

## Statistical information and image preparation

ImageJ software (from NIH) was used to quantify all the immunoblots and EMSA blots. GraphPad Prism software (version 9.3.1) was used to prepare all graphs, calculate $K_d$, IC₅₀, and EC₅₀ values, and to perform statistical tests. Statistical significance was determined using unpaired two-sided Student's *t*-tests, and $p < 0.05$ was considered statistically significant. All values are represented as mean ± SD. FlowJo (v.10) was used to analyze flow cytometric data. Final figures were prepared using Illustrator (Adobe).

## Reporting summary

Further information on research design is available in the Nature Portfolio Reporting Summary linked to this article.

## Data availability

All data supporting the findings of this study are available within the paper and its Supplementary Information. The mass spectrometric-based data generated in this study has been deposited in the ProteomeXchange Consortium-PRIDE repository database under accession code PXD042808. All the flow cytometric data can be found in the Source data files folder. All raw data can be found in the Source file (excel) in the Source data files folder. The AlphaFold structure of *Tb*RPA1 can be found at https://alphafold.ebi.ac.uk/entry/Q384B5. Information on the *Tb*RPA1 gene can be found at https://tritrypdb.org/tritrypdb/app. Source data are provided with this paper.

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

## Acknowledgements

We would like to thank David Galler and Diana Rodriguez (Queens College) for the synthesis of early-stage RPA1 inhibitors; Hemraj Rimal and Saroj Biswas (Rutgers University) for help with some of the pSR plasmid construction; Olivia Haskin (Rockefeller University, High School Outreach Program) for making the pOH10 vector; Carolyn Suzuki, Venkatesh Sundararajan and Sakthijothi Muthu at Rutgers University for help with human cell culturing; Abhiruchi Kant (Rutgers University) for help with SEC; Matthew Neiditch and Carolyn Suzuki (Rutgers University) for training and sharing the MST instrument; Bibo Li at Cleveland State University for anti-RPA1 antibodies; Natalia Ketaren (Rockefeller University) for Mass photometry; PHRI Flow cytometry core (Rutgers University); Proteomic core at Rutgers University; and Life Science Editors for help with manuscript editing. This work was supported by the National Institute of Allergy and Infectious Diseases of the National Institutes of Health [grant number R01AI127562 to H.-S.K.] and PHRI-Rutgers start-up funds to H.-S.K. This work was also supported by the National Institute of General Medical Sciences of the National Institutes of Health under award number SC2GM130470 to J.Y.C.

## Author contributions

H.-S.K and J.Y.C. conceived the concept of the study and supervised the study. A.M. carried out all the experiments and prepared the initial draft of the manuscript. E.E. made constructs to generate T. brucei AMT2 cell line and Tc & LmexDBD-AB proteins. Z.H. synthesized all JC compounds. S.M. synthesized ATRIP. H.-S.K., J.Y.C., A.M., and E.E. wrote and revised the manuscript.

## Competing interests

The City University of New York, Rutgers, The State University of New Jersey, and Instituto de Investigaciones Biotecnológicas -Instituto Tecnologico de Chascomus Consejo de Investigaciones Cientificas y Tecnicas, Universidad Nacional De San Martin have filed a patent application relating to JC-229 in this work for the treatment of *Trypanosoma brucei* (patent application no.63428794 pending), with J.Y.C., Z.H., H.-S.K., A.M., and E.E. named as inventor. The remaining authors declare no competing interests.
