## [Peer Review File · Nature Communications]

Identification of a small-molecule inhibitor that selectively blocks DNA-binding by Trypanosoma brucei Replication Protein A1REVIEWER COMMENTS

Reviewer #1 (Remarks to the Author):

1) The manuscript my Kim and colleagues describes the discovery of a novel small molecule inhibitor of the *T. brucei* single-stranded DNA binding protein RPA. The molecule is chemically similar to hsRPA DBD-F inhibitors discovered by other investigators. The manuscript contains a mix of compelling and not so compelling experiments. The activity of JC-229 against the *T. brucei* RPA is clear and convincing and the combination of EMSA and MST provide important information. The experiments describing cellular activity against *T. brucei* are also well designed executed and interpreted. The limitation comes from the description of JC-229 and its origins, the analysis of specificity versus hsRPA and activity in human cancer cells.

2) The description as to how JC-229 was identified is lacking. The references to figure 1D and supplemental figure 2 are not sufficient to described how 229 was identified. Clearly JC-229 is based on analogs developed in the Fesik lab targeting the DBD-F. The paragraph starting at line 162 should be expanded to include the ref 24 and (compounds 3 and 4 therein) as the starting point. It is also not clear where the analogs that were tested came from. Was it an SAR effort with novel synthesis and JC-229 was the best? Were analogs purchased to ID relevant pharmacophores?

3) The conclusion that JC-229 is specific for *T. brucei* and not human RPA is not well supported. The class of inhibitors being tested are know not to inhibit RPA-DNA binding activity so to test JC-229 is largely irrelevant. The assay that needs to be performed is assessing the impact of JC-229 on a potential interaction with the hsRPA DBD-F and measuring a relevant protein-protein interaction. The hsRPA EMSAs and MST data presented in figures 4 and 5 are not necessarily relevant and do little to advance our understanding of the specificity of JC-229.

4) In light of the above comment, the finding that that there is some cytotoxic activity against human cancer cells suggest that there is an effect on some protein, maybe RPA, making the analysis of DBD-F interaction even more relevant.

5) The differential sensitivity of the human cells to *T. brucei* cannot be explained based on the "selective versus the RPA's. This conclusion needs to be tempered in the manuscript.

6) Specific against *L. mex* RPA is interesting in the context of implication of ser105, However, definitive demonstration of on-target activity against TbRPA being responsible for the cellular phenotype observed would require creation of the S105T mutant in *T. Brucei* and demonstrate reduced activity. The data in figure 3 showing that JC229 mimics RPA depletion, is a =n excellent correlation, but not definitive proof.

6) The authors state the lack of the DBD-F in *T. brucei* is an opportunity for selective therapeutic development. Can the authors explain this statement and how the lack of a domain may help in developing selective inhibitors for the *T. brucei*?

Reviewer #2 (Remarks to the Author):

In this manuscript, Mukherjee and colleagues present a highly specific inhibitor for the *Trypanosoma brucei* RPA complex (JC-229), which was selected based on structure modeling of TbRPA. The authors showed that JC-229 can kill parasites with a reasonably low EC50. Furthermore, the authors show that JC-229-treated cells developed phenotypes comparable to TbRPA1-depleted parasites, including an impressive complete block of DNA synthesis 24 hours post treatment. Inhibition of DNA binding was studied using EMSA and MST, which confirmed inhibition of TbRPA1 but not of homologs from *humas*, *T. cruzi* and *Leishmania*. Amino acid exchange within RPA1 DBD-A identified S105 to be involved in JC-229 interaction, which might give some hints for future design of JC-229 derivatives.

This manuscript is very concise and well-written. A pleasure to read. All experiments were performed with a high standard and I have (surprisingly) nothing I could suggest to improve them. The conclusions and claims are adequate, problems and gaps, which need to be addressed in the future are discussed. There are only a few minor issues (see below).

Line 44-45: "These drugs have high rates of toxicity, are difficult to administer, and drug resistance quickly arises." This is not true for all drugs mentioned above and should be stated in a more differentiated manner.

The first paragraph of the result section is somewhat confusing. The authors claim that DBD-F is absent in TbRPA1 and that the tertiary structure of DBD-F resembles that of TbRPA1 DBD-A, which I cannot really retrace from figure 1B.

Line 158: "... TbRPA1 DBD-A binds ssDNA" I guess this is only a prediction based on the structure modeling.

Line 159: What is DBD-AB? This should be introduced earlier.

Fig 1D (synthesis scheme) doesn't match the text of the manuscript (analogs of HsRPA1 inhibitors)

Line 222: The authors state that JC-229 has selective toxicity against *T. brucei*. This should be described more carefully because the toxicity of JC-229 for eg. HEK cells is quite high (LC50 18 μ M).

Reviewer #3 (Remarks to the Author):

Mukherjee et. al. describe characterization of a small molecule inhibitor of RPA (JC-229) with reasonable selectivity. The logic here is to use previously known inhibitors of RPA and in silico docking to identify compounds to target *Trypanosoma brucei* RPA. They show that JC-229 blocks the DNA binding activity of Tb-RPA, but does not affect human RPA. In vivo, the compound inhibits Tb cell growth, but is mildly toxic to human cells. Thus, the authors provide evidence that targeting Tb RPA might be an effective therapeutic strategy to treat African trypanosomiasis.

Current compounds targeting RPA are notoriously non-specific and have never made it past preliminary screens. Even through groups such as Fesik's have investigated the possibility, the compounds never manifested into anything useful. Thus, targeting RPA (especially the OB domains) to treat trypanosomiasis (while an interesting option) will likely not be a viable therapeutic strategy. Major Comments:

While the experiments are done with reasonable clarity, the premise and the foundational work is quite weak. For example, attempting to drug a protein without adequately characterizing its biophysical properties and establish a mechanism of action is not appropriate. The authors cite a paper from 2016 (Pavani et. al PLoS Negl Trop Dis 10,953 e0005181) as the basis to state that only DBD-A binds to DNA. A closer look at this paper reveals too many poorly done experiments. At a minimum, this work needs to be more rigorously investigated before any conclusions can be made.

If findings from the Pavani paper are indeed true, then RPA from Tb cannot be considered to function canonically as RPA from other eukaryotes. In fact, alignment of RPA70 with human and yeast RPA show a very high degree of conservation and thus, multiple DNA binding domains must exist. Something else is amiss in that study and it should not be used as a rubric.

However, I cannot find fault with the actual data shown in this manuscript. The authors capture stoichiometric binding of both TbRPA and hRPA. This shows that their biochemical purifications of the proteins are active. I would like to see the following experiments before this study can be evaluated further:

1. What is the oligomeric state of TbRPA? Authors can use SEC, AUC, or Mass Photometry to establish this.
2. What is the site-size for TbRPA? Authors can use the intrinsic Trp fluorescence signal and quenching

upon ssDNA binding to measure the site-size. This experiment is important in guiding the choice of ssDNA length used for the experiments. For example, in experiments where just the AB domains are used, a shorter ssDNA should be used as multiple AB molecules will bind. In the EMSA experiments, their experiments show that the compound blocks AB oligomerization for hRPA. Using a shorter length ssDNA will circumvent these issues. This also might explain why the IC50 for the compound while targeting AB versus full length are different.

3. Is there a reason why DNA binding experiments are done with 10 mM NaCl?

4. Can the authors compare the SWISS-MODEL versus AlphaFold model of TbRPA1. Do the conclusions hold?

5. In the AlphaFold model, the OB-domains of A and B are very similar. Is there a reason why the authors believe the compound to be specific for OB-A in Tb?

6. Since the AlphaFold model is a reasonable guide to make truncations, can the authors generate OB-A and OB-B from TbRPA1 and test binding of the compound?

7. The biggest concern here is that a compound that supposedly blocks DBD-A is shown to inhibit the ssDNA binding activity of full length TbRPA. In fact, this assumption is incorrect for human RPA. When DBD-A and DBD-B are removed, the trimerization core retains DNA binding activity. In fact, in yeast RPA, the trimerization core provides better stability for ssDNA interactions compared to DBD-A. Thus, the model here predicting that JCC-229 interacts selectively with DBD-A and induces this global inhibition of ssDNA binding by TbRPA does not agree with other recent biochemical investigations.

8. However, if the data shown in the paper are taken at face value, another interpretation would be that the compound is binding to multiple domains, or regions that induce allosteric changes in TbRPA1. Given that most of the interaction data is speculative or derived from investigation of either DBD-A or DBD-AB, the authors should be a bit more circumspect in their interpretation.

Minor Comments:

1. Abstract line 1: complex is repeated twice.

2. Line 90: The statement that there are 6 DBDs in RPA is incorrect. There are 6 OB domains and four of these primarily coordinate DNA binding and are called DBDs.

3. Line 94: The statement that DBDs A and B 'dominate' during ssDNA is being revised through new studies looking at the dynamics of RPA domains. Please refer to more current literature on RPA dynamics.

4. Line 111: Shouldn't drug discovery follow a thorough characterization of the target enzyme? How can you assess drug efficacy and mechanism of action without knowledge of the enzyme in question.

5. Line 114: DBD-A of eukaryotic RPA is the major ssDNA binding domain. This statement is inaccurate. DBDs A and B are more dynamic compared to the Trimerization core of RPA.

6. Line 125: There is so many contradictions in the introduction and the logic for this paper. The authors state that the drug discovery effort provides a unique opportunity as the Tb RPA does not possess a F-domain. However, they are targeting DBD-A. This is present in human RPA as well. How does their approach involve an advantage by targeting DBD-A?

7. Line 131: What is an in-house 3D structural model? Just state SWISS-MODEL or use the new AlphaFold Model (which looks quite reasonable)

8. A 2 to 5-fold difference in affinity is not an ideal drug target.

9. Line 383: 'Binding pose' is not the correct terminology

Response to reviewers

We are very grateful for the reviewers' helpful comments – we believe that addressing them has improved the manuscript substantially.

We have added an additional author, Shuai Ma, as he synthesized ATRIP peptide that we used to test human DBD-F interaction with ATRIP and the effect of JC-229 on their interaction *in vitro*.

In this revision, as reviewers suggested, we have characterized biochemical properties more in detail, performed genetic studies on trypanosome mutants expressing the S105T mutant RPA1 & investigated the effect of JC-229 on human DBD-F's PPI function. Data from these experiments are presented in one additional main figure and 10 additional supplementary figures. Reviewers' comments are written in italic font 11 and our responses are in regular font 11 followed by an arrow (indented). To assist with reviewing, we have marked the changes in blue font in our revised version. Please see below our point-by-point response to each of reviewers' comments/concerns.

Reviewer #1 (Remarks to the Author):

*1) The manuscript by Kim and colleagues describes the discovery of a novel small molecule inhibitor of the *T. brucei* single-stranded DNA binding protein RPA. The molecule is chemically similar to hsRPA DBD-F inhibitors discovered by other investigators. The manuscript contains a mix of compelling and not so compelling experiments. The activity of JC-229 against the *T. brucei* RPA is clear and convincing and the combination of EMSA and MST provide important information. The experiments describing cellular activity against *T. brucei* are also well designed executed and interpreted. The limitation comes from the description of JC-229 and its origins, the analysis of specificity versus hsRPA and activity in human cancer cells.*

- Thank you very much for your kind comments. We have revised the introduction and the first subsection of results (page 7, line 166-174), which now include description of the origin of JC-229. Please see our response to your comment 2 for detail.

Regarding “the analysis of specificity versus HsRPA and activity in human cancer cells”: Although the inhibition was weaker than to *T. brucei* cells, JC-229 inhibits the growth of HeLa and HEK293 cells (EC₅₀ of 32 and 18μM respectively, Figure 3f). So, it is possible that either JC-229 inhibits the PPI function of human RPA1 or that there are other targets for JC-229 in human cells. As the reviewer suggested, we examined the interaction of human DBD-F with peptides of ATRIP, one of the interacting proteins of human RPA1 N-terminal domain. In this *in-vitro* experiment, we found that recombinant DBD-F interacts with ATRIP peptide and the interaction is inhibited by JC-229, but with a high IC₅₀ value, ~60μM (the new data is now presented in **Supplementary Figure 14** and ATRIP peptide synthesis scheme in **Supplementary Figure 15**). Thus, we conclude that JC-229 does not inhibit the ssDNA-binding activity of human RPA1 DBD-AB or RPA complex (Figure 4 and 5), and does not significantly inhibit the ATRIP-DBD-F interaction. However, because there are other DDR proteins that interact with human RPA1 DBD-F, our result does not exclude a possibility that JC-229 may inhibit interaction between DBD-F and other DDR factors, which could cause accumulative inhibitory effects. Or simply JC-229 may have other targets that contain an OB fold structure.

2) The description as to how JC-229 was identified is lacking. The references to figure 1D and supplemental figure 2 are not sufficient to describe how 229 was identified. Clearly JC-229 is based on analogs developed in the Fesik lab targeting the DBD-F. The paragraph starting at line 162 should be expanded to include the ref 24 and (compounds 3 and 4 therein) as the starting point. It is also not clear where the analogs that were tested came from. Was it an SAR effort with novel synthesis and JC-229 was the best? Were analogs purchased to ID relevant pharmacophores?

→ We apologize for the lack of information on JC-229 in our first manuscript. In our revised manuscript, we have described the origin of JC-229 (page 7, line 166-174) and also included data obtained from additional compounds that we tested (these compounds were published previously from Fesik's laboratory). Briefly, based on the results of the structure alignment of *HsDBD-F* and *TbDBD-A* (**Fig 1b**), we hypothesized that chemical agents targeting the *HsDBD-F* could inhibit the ssDNA-binding of *TbDBD-A*. As a proof of principle study, we chose few known *HsRPA1* inhibitors (which were reported to bind to the *HsDBD-F*) for synthesis and tests in *T. brucei*. A triazole compound (compound 19o in ref 31) and diphenylpyrazoles (such as 5f, 5g, and 4f in ref 33) were synthesized in our lab, which were renamed as JC-230, JC-231, JC-232, and JC-233, respectively. They did not show toxicity to *T. brucei* cells and these results are now included in Supplementary Figure 2.

JC-229 is an analog of compounds 3 and 4 (Ref 20: this was the ref 24 in our first manuscript). We chose to synthesize and test JC-229 rather than compound 3 or 4, because it was easily accessible by synthesis. We were also concerned about the *T. brucei* membrane permeability of compounds 3 and 4 since there are three hydrogen-bond donors in amide, sulfonamide, and carboxylic acid functional groups in these compounds. The presence of multiple hydrogen bond donors can cause membrane permeability issues in human cells. In addition, the carboxylic acid moiety of compound 3 or 4 could be oriented to the solvent accessible area without any interactions with *TbDBD-A*, according to our docking model (Fig 1c). Thus, we synthesized JC-229 for tests in *T. brucei* cell growth.

After the synthesis of these five compounds, we tested their effects on *T. brucei* cell growth and obtained EC₅₀ of 6μM for JC-229 and 50μM for JC-230. JC-231~233 were not toxic to *T. brucei* cells. We also tested TDRL-505 (commercially available) and found that TDRL-505 did not show any toxicity even at 60 μM concentration (**Supplementary Figure 2**). We were unable to test HAMNO and NSC15520 because we could not dissolve these in *T. brucei* media.

Due to the limits of word count, we did not describe details of the origin of JC-229 and other JC compounds in the first manuscript. As described above, JC-229 came from synthesis and assays of the five compounds, not from SAR efforts nor purchase of agents based on relevant pharmacophores. Since the revised manuscript includes many new results (11 new figures total) from additional experiments, the origin of JC-229 is briefly described in the results section.

3) *The conclusion that JC-229 is specific for T. brucei and not human RPA is not well supported. The class of inhibitors being tested are known not to inhibit RPA-DNA binding activity so to test JC-229 is largely irrelevant. The assay that needs to be performed is assessing the impact of JC-229 on a potential interaction with the hsRPA DBD-F and measuring a relevant protein-protein interaction. The hsRPA EMSAs and MST data presented in figures 4 and 5 are not necessarily relevant and do little to advance our understanding of the specificity of JC-229.*

→ We agree with the reviewer! JC-229's effect on *HsDBD-F* needs to be tested, as JC-229 inhibited human cancer cell growth and JC-229 is in fact an analog of a PPI inhibitor. To answer this question, we generated 6xHis-tagged *HsDBD-F* expression vectors and purified *HsDBD-F* protein by Ni-NTA. We synthesized ATRIP peptide and labeled with Cy5 at the N-terminus. We then performed MST to examine *HsDBD-F* binding with ATRIP, and JC-229's inhibition of this interaction. Data is now presented in **Supplementary Figure 14**. We found that *HsDBD-F* interacts with ATRIP peptide with $K_d \sim 34 \mu\text{M}$ and JC-229 inhibits this interaction with IC₅₀ $\sim 60 \mu\text{M}$. The EC₅₀ values for HeLa and HEK293 were $\sim 32 \mu\text{M}$ and $18 \mu\text{M}$, respectively. Thus, it is possible that the inhibition of *HsDBD-F* by JC-229 might cause the inhibition of HeLa and HEK293 cell growth or that JC-229 may have other target proteins, perhaps other OB-fold containing proteins. We revised the text accordingly.

JC-229 is more active on the inhibition of *Hs*DBD-F than *Hs*DBD-AB, as we were able to detect some inhibition of DBD-F's binding with ATRIP peptide (~60 μ M) (**Supplementary Figure 14**) but did not detect any measurable inhibition of ssDNA-binding activity of *Hs*RPA1 DBD-AB or *Hs*RPA complex (Figure 4 and 5). As predicted by molecular modeling, JC-229 was highly effective on the inhibition of ssDNA binding of *T. brucei* RPA and DBD-AB. Overall, our data indicate that JC-229 can really differentiate OB-fold structures in *Hs*DBD-F, *Hs*DBD-A, and *Tb*DBD-A. The interaction of JC-229 with *Tb*RPA will be further studied using X-ray crystallography in the near future.

4) *In light of the above comment, the finding that there is some cytotoxic activity against human cancer cells suggest that there is an effect on some protein, maybe RPA, making the analysis of DBD-F interaction even more relevant.*

→ We agree! We address this concern in (3) and also in (1).

5) *The differential sensitivity of the human cells to T brucei cannot be explained based on the “selective versus the RPA’s. This conclusion needs to be tempered in the manuscript.*

→ We agree. The differential sensitivity cannot be explained by selectivity only, as there are many factors that can affect toxicity of drugs, for example, drug delivery to the nucleus. We rephrased the sentence accordingly throughout the manuscript.

6) *Specific against L. mex RPA is interesting in the context of implication of ser105, However, definitive demonstration of on-target activity against TbRPA being responsible for the cellular phenotype observed would require creation of the S105T mutant in T Brucei and demonstrate reduced activity. The data in figure 3 showing that JC229 mimics RPA depletion, is a =n excellent correlation, but not definitive proof.*

→ We agree! If *Tb*RPA1 is truly the major target of JC-229 and the S105T mutation blocks the JC-229 docking at DBD-A of *Tb*RPA1, trypanosome cells expressing the mutant RPA1 should be resistant to JC-229. As the reviewer pointed out, this would require generation of new *T. brucei* cell lines and plasmids for knockout and knock-in. Thanks to the editor and reviewers who gave us a few months of extension, we were able to generate all necessary plasmids & cell lines and examine the resistance of *T. brucei* S105T mutants to JC-229. We tested three independent clones of S105T mutant trypanosomes and found that all three were resistant to JC-229 with following EC₅₀ values: 5.7 μ M for WT, and 5 μ M for single KO (WT/ Δ) and 30.4, 21.7, and 19.6 μ M for three S105T mutant clones. **These data are now shown in new Figure 8 and Figure 8d is attached.**

We have also performed another *in vitro* experiment with S105T mutant protein, although this was not requested by the reviewers. Reviewer 3 was concerned whether *Tb*DBD-A would be the sole target domain of JC-229 and whether other domains (perhaps DBD-C, D, and/or E, a trimerization core) might be targeted by this compound. Because we used *Tb*DBD-AB for ssDNA-binding assay, we could not definitively confirm that DBD-A is the only target of JC-229. *Tb*DBD-AB containing S105T mutation completely lost the inhibition by JC-229, suggesting that within *Tb*DBD-AB fragment, DBD-A has the major target site for JC-229. However, this assay does not rule out the possibility that JC-229 might inhibit the activity of DBD-C or D. To see whether DBD-A of *Tb*RPA1 is indeed the major (sole) target of JC-229, we purified *Tb*RPA complex containing S105T and performed EMSA and MST. We found that the mutant protein complex retained ssDNA-binding activity as the level of WT complex but the mutant's ssDNA-binding activity was unaffected by JC-229 treatment, unlike the WT. Our new data confirms that *Tb*RPA1 DBD-A is the major target of JC-229. These data are now shown in **Supplementary Figure 18**.

6) *The authors state the lack of the DBD-F in T brucei is an opportunity for selective therapeutic development. Can the authors explain this statement and how the lack of a domain may help in developing selective inhibitors for the T-brucei?*

- Thank you for the comment! We believe that the reviewer is referring to this sentence in page 5, lines 124-126 in our first version of the manuscript. “Although available data are limited, unique features of RPA1 in these parasites, such as its nuclear export during differentiation and roles in telomere protection, offer opportunities for selective therapeutic development^{35-38,40,41}.”. Reviewer 3 also pointed this out in his/her minor concern 6. We realized that this was not clear and even confusing. To avoid any confusion, we removed “the lack of DBD-F in RPA1”. Please see our response to the reviewer 3’s minor concern 6.

Reviewer #2 (Remarks to the Author):

In this manuscript, Mukherjee and colleagues present a highly specific inhibitor for the Trypanosoma brucei RPA complex (JC-229), which was selected based on structure modeling of TbRPA. The authors showed that JC-229 can kill parasites with a reasonably low EC50. Furthermore, the authors show that JC-229-treated cells developed phenotypes comparable to TbRPA1-depleted parasites, including an impressive complete block of DNA synthesis 24 hours post treatment. Inhibition of DNA binding was studied using EMSA and MST, which confirmed inhibition of TbRPA1 but not of homologs from humas, T. cruzi and Leishmania. Amino acid exchange within RPA1 DBD-A identified S105 to be involved in JC-229 interaction, which might give some hints for future design of JC-229 derivates.

This manuscript is very concise and well-written. A pleasure to read. All experiments were performed with a high standard and I have (surprisingly) nothing I could suggest to improve them. The conclusions and claims are adequate, problems and gaps, which need to be addressed in the future are discussed. There are only a few minor issues (see below).

- Thank you very much for your kind comments!

Line 44-45: “These drugs have high rates of toxicity, are difficult to administer, and drug resistance quickly arises.” This is not true for all drugs mentioned above and should be stated in a more differentiated manner.

- We modified the sentence to “However, melarsoprol is highly toxic and eflornithine is difficult to administer and costly, and many including melarsoprol, suramin, and pentamidine have problems with drug resistance^{2,3}. There are no good treatment options for nagana.”
(page 3, lines 44-47)

The first paragraph of the result section is somewhat confusing. The authors claim that DBD-F is absent in TbRPA1 and that the tertiary structure of DBD-F resembles that of TbRPA1 DBD-A, which I cannot really retrace from figure 1B.

- We apologize for the confusion. To avoid confusion, we rearranged the text and in our revised manuscript, we emphasized that all DBDs in RPA complex including the DBD-F have an OB-fold structure. Since there are some structural similarities between *HsDBD-F* and *HsDBD-A, B, and C* of RPA1 (basically all have an OB fold), we postulated that there should be some structural similarities between *HsDBD-F* and *TbRPA1 DBD-A*, which also has an OB-fold structure in the 3D model (both SWISS-MODEL and AlphaFold).

We realized that it might be difficult to see the resemblance. Below structure has been added to **the main Figure 1b**. In this image, the *HsRPA1 DBD-F* structure (blue) is overlaid to that of *TbRPA1 DBD-A* (red) by Pymol software, and it shows that both *HsDBD-F* and *TbDBD-A* domains have similar tertiary structures with aligned five beta-sheet strands where an interacting protein and potentially ssDNA binds, respectively. For clarity, we have added additional structure models in **Supplementary Figure 1** (Please also see our response to the reviewer 3's comments 4 and 5).

Line 158: "... *TbRPA1 DBD-A* binds ssDNA" I guess this is only a prediction based on the structure modeling.

- Yes, it is based on the 3D structure from SWISS-MODEL and also from AlphaFold, which is added in the revised version (**Supplementary Figure 1b-e**).

Line 159: What is DBD-AB? This should be introduced earlier.

- It is RPA1 DBD-A and B domain in short. We have introduced DBD-AB earlier, as suggested (page 4, line 95).

Fig 1D (synthesis scheme) doesn't match the text of the manuscript (analogs of *HsRPA1* inhibitors)

- The synthesis scheme is correct. JC-229 is an analog of *HsRPA1* inhibitors developed in the Fesik laboratory but JC-229 has not been synthesized and tested (or reported) by his lab. We synthesized JC-229 by following the synthetic scheme in Figure 1d and its structure was confirmed by ¹H NMR, ¹³C NMR (Supplementary Figure 3). The synthetic scheme is matched with the procedures in the method section, which is different from the method used for compound 3 or 4 in the ref 20.

Line 222: The authors state that JC-229 has selective toxicity against *T. brucei*. This should be described more carefully because the toxicity of JC-229 for eg. HEK cells is quite high (LC50 18 μM).

- We have revised "selective toxicity" to "higher toxicity against *T. brucei* than to human cells" (page 8, line 224-225). We realized that we have used "selective"/ "selectivity" elsewhere in the manuscript. We checked thoroughly and revised them where we might have overinterpreted.

Reviewer #3 (Remarks to the Author):

Mukherjee et. Al. describe characterization of a small molecule inhibitor of RPA (JC-229) with reasonable selectivity. The logic here is to use previously known inhibitors of RPA and in silico docking to identify compounds to target Trypanosoma brucei RPA. They show that JC-229 blocks the DNA binding activity of Tb-RPA, but does not affect human RPA. In vivo, the compound inhibits Tb cell growth, but is mildly toxic to human cells. Thus, the authors provide evidence that targeting Tb RPA might be an effective therapeutic strategy to treat African trypanosomiasis.

Current compounds targeting RPA are notoriously non-specific and have never made it past preliminary screens. Even through groups such as Fesik's have investigated the possibility, the compounds never manifested into anything useful. Thus, targeting RPA (especially the OB domains) to treat trypanosomiasis (while an interesting option) will likely not be a viable therapeutic strategy.

→ Thank you very much for your kind comments and concerns!

Regarding the concern whether *TbRPA1* would be a good target for development of anti-trypanosomal compound:

Because *T. brucei* (and related kinetoplastids, *T. cruzi* and *Leishmania*) is evolutionarily divergent from its human host, even the highly conserved replication proteins have poor sequence identities. If a target protein in the pathogen has unique features differentiating it from its host counterpart, while keeping the same essential function, those features can be useful in developing selective inhibitors against the pathogen protein. We believe that our results provide valuable information for future studies on anti-trypanosome therapy development.

Major Comments:

*While the experiments are done with reasonable clarity, the premise and the foundational work is quite weak. For example, attempting to drug a protein without adequately characterizing its biophysical properties and establish a mechanism of action is not appropriate. The authors cite a paper from 2016 (Pavani et al. *PLoS Negl Trop Dis* 10,953 e0005181) as the basis to state that only DBD-A binds to DNA. A closer look at this paper reveals too many poorly done experiments. At a minimum, this work needs to be more rigorously investigated before any conclusions can be made.*

*If findings from the Pavani paper are indeed true, then RPA from *Tb* cannot be considered to function canonically as RPA from other eukaryotes. In fact, alignment of RPA70 with human and yeast RPA show a very high degree of conservation and thus, multiple DNA binding domains must exist. Something else is amiss in that study and it should not be used as a rubric.*

*However, I cannot find fault with the actual data shown in this manuscript. The authors capture stoichiometric binding of both *TbRPA* and *hRPA*. This shows that their biochemical purifications of the proteins are active. I would like to see the following experiments before this study can be evaluated further:*

→ Thank you for your overall feedback! Because of the large evolutionary distance between trypanosomes and human, it is not uncommon to have poor sequence identities and similarities between orthologs, even for the highly conserved replication proteins. However, the sequence similarities among *TbRPA1* and *HsRPA1*, also RPA1 from other organisms are considerably high, compared to other proteins that are less conserved in *T. brucei*. We and a few labs have studied the recruitment and localization of *TbRPA1* or *TbRPA2* to the sites of DNA damage induced by IR, HU, or I-Sce-induced double strand break (Glover et al, *mBio*, 2019; PA Marin et al, *Scientific Reports*, 2018; JA Black et al, *Cell Rep*, 2020). Our lab also detected *TbRPA1* foci formation upon DNA damage and in replication-defective mutant cells (Kim *NAR*, 2019). *TcRPA1* and *Leishmania* RPA1 with ~80% sequence identities have also been studied in Elias lab and Cano lab.

Although there is still much to learn about these proteins compared to human and yeast models, the fact that *TbRPA* functions in DNA damage response (DDR) in *T. brucei* suggests that further investigation into its genetic, molecular, and biochemical properties, including its PPI function, is warranted. While our work represents the first biochemical and genetic studies on *TbRPA1*, we recognize that more research is needed to fully understand the intricacies of DNA replication and repair in trypanosomatids. We will continue to conduct our research on this topic.

We appreciate the reviewer's comments and we hope that we made significant improvements to our manuscript based on their feedback. We conducted several additional experiments to thoroughly characterize the properties of *TbRPA* protein, as suggested by the reviewer. These include: (i) *T. brucei* RPA binding with ssDNA probe with different lengths using EMSA and MST. JC-229 inhibition of these binding was also examined. These data are presented in **Supplementary Figure 10 and 11**. (ii) We investigated the ssDNA-binding properties of *TbDBD-A* and *TbDBD-B* separately (**Supplementary Figure 13**). (iii) We determined the binding property of mutant *TbRPA* complex containing the S105T mutation, with or without JC-229 treatment (**Supplementary Figure 18**). (iv) We confirmed the presence of *TbRPA2* and *TbRPA3* in 6xHis-*TbRPA1* Ni-NTA pulldown by Mass Spectrometry (**Supplementary Figure 7**) and examined the oligomerization state of *TbRPA* complex by SEC and MP (**Supplementary Figure 8**). Results obtained from these experiments further expanded our knowledge on *TbRPA* biochemical properties and led us to discover the non-canonical features of *TbRPA1*, some of which are quite different even from *T. cruzi* RPA1. Please see below our point-by-point response to the comments.

1. *What is the oligomeric state of TbRPA? Authors can use SEC, AUC, or Mass Photometry to establish this.*

- ➔ Because 6xHis is tagged only at the *TbRPA1* subunit and not *TbRPA2* and *TbRPA3* subunits, these smaller subunits will not be co-purified with 6xHis-*TbRPA1* unless they form a complex with 6xHis-*TbRPA1*. To confirm that all three subunits are present in the 6xHis pulldown fraction by Ni²⁺-NTA resin, we analyzed the purified fraction by mass spectrometry and confirmed that all three subunits are present (**Supplementary Figure 7**).

To determine the oligomeric state of *TbRPA* complex, we performed Size Exclusion Chromatography (SEC) and Mass Photometry (MP) (**Supplementary Figure 8**), as suggested by the reviewer. *TbRPA* complex was eluted at 14.7ml, which corresponds to ~142 kDa (calculated from standard curve), which is larger than the predicted *TbRPA* complex of 94 kDa = 52 kDa (*TbRPA1*) + 28 kDa (*TbRPA2*) + 14 kDa (*TbRPA3*). Because an asymmetric molecule will elute with a higher molecular weight compared to a globular one (Fágáin et al, *Protein Chromatography*, 2016), it is possible that *TbRPA* is in a trimeric state but elute earlier due to their elongated shape. We used another method to determine the oligomeric state of *TbRPA*, Mass Photometry. We found a major peak corresponding to 91 kDa, which is close to the expected molecular weight of *TbRPA* full complex. These experiments (SEC, MP & Mass spec) confirm that our purified fraction has *TbRPA* trimer.

2. *What is the site-size for TbRPA? Authors can use the intrinsic Trp fluorescence signal and quenching upon ssDNA binding to measure the site-size. This experiment is important in guiding the choice of ssDNA length used for the experiments. For example, in experiments where just the AB domains are used, a shorter ssDNA should be used as multiple AB molecules will bind. In the EMSA experiments, their experiments show that the compound blocks AB oligomerization for hRPA. Using a shorter length ssDNA will circumvent these issues. This also might explain why the IC50 for the compound while targeting AB versus full length are different.*

- ➔ Thank you for the comment!! We used dT₃₂ oligo because this is commonly used to examine ssDNA-binding activity of RPA and DBD-AB in humans and yeasts (Shuck et al, *Cancer Res*, 2010; Yates et al, *Nat Commun*, 2018). We assumed that about three molecules of *TbDBD-AB* would occupy dT₃₂, while one molecule of *TbRPA* complex would bind dT₃₂, based on the human and yeast RPA studies (Bochkareva et al, *EMBO J*, 2002; Dueva Iliakis et al, *NAR cancer*, 2020). It turned out it is a little different with *T. brucei* RPA proteins. We have examined *TbRPA1* DBD-AB and *TbRPA* complex's binding with shorter oligos, dT₂₀ and dT₁₂, and also examined whether JC-229's inhibitory effect changes depending on the length of ssDNA using EMSA and MST (**Supplementary Figure 10 & 11**).

As we expected, *TbRPA* complex binds dT₂₀ probes as well as dT₃₂. K_d values were 15.5 nM for dT₃₂ and 17.2 nM for dT₂₀. JC-229 inhibits the interaction of *TbRPA* with dT₃₂ or dT₂₀ with similar IC₅₀ (~33.9 nM). For the binding with dT₁₂, the MST values fluctuated frequently, reflecting rapid association and dissociation of *TbRPA* on dT₁₂ as seen in other eukaryotes (Wang et al, *JBC*, 2019). This indicates that ~20 nt is the minimal length that *TbRPA* complex can stably bind (site size for *TbRPA* complex). So, 20-mer oligo can accommodate one RPA full-length complex in *T. brucei*, while that size can accommodate only the DBD-A, B and C of human and yeast RPA (Bochkareva et al, *EMBO J*, 2002; Kumaran et al, *Biochemistry*, 2006).

We expected that *TbDBD-AB* will interact well with all oligo substrates (dT₃₂, dT₂₀, and dT₁₂), but less JC-229 will be required to inhibit the interaction with dT₂₀ or dT₁₂, compared to dT₃₂. In our new experiments, we discovered that *TbDBD-AB* interacts with dT₂₀ almost as well as with dT₃₂ (K_d values of 30.5 nM vs. 24.6 nM). But surprisingly, a much higher concentration of JC-229 was needed to block *TbDBD-AB* interaction with dT₂₀ (IC₅₀ of 881 nM for dT₂₀ vs. 228 nM for dT₃₂). We did not detect a strong interaction between *TbDBD-AB* and dT₁₂. We were able to detect some binding with dT₁₂ in MST (K_d value ~60.5 nM) but were unable to detect any by EMSA experiment, indicating that *TbDBD-AB* needs at least 20 nt for stable interaction with ssDNA.

A closer look at the biochemical properties with additional experiments led us to discover novel features of *TbRPA1*, which could have been overlooked. We really appreciate the reviewer 3 for this. In higher eukaryotes, as small as 8 to 12-mer oligo can form stable complex with the AB domain whereas it takes at least 30 nucleotides for formation of a stable complex with the full-length complex. Interestingly, our assays suggested that as small as 20-mer can form a stable complex with both AB domain and RPA complex in *T. brucei*. There are clear differences between *T. brucei* RPA and RPA in the host, which could provide valuable insights on developing inhibitors that target pathogen-specific features for selective killing.

Regarding the reviewer's comment "their experiments show that the compound blocks AB oligomerization for hRPA": We did not show this.

3. Is there a reason why DNA binding experiments are done with 10 mM NaCl?

- We used 10mM NaCl in the DNA binding experiments, as reported by Oakley group (Glanzer et al, *J Antimicrob Chemother*, 2016). This reference has been added (Ref. 70). While optimizing our EMSA binding buffer composition for 5'IRDye800-labelled oligos, we slightly modified their protocol by adding 10 mM MgCl₂ and 0.1 mg/ml BSA in the buffer. The buffer composition was kept constant for all DNA binding assays to ensure the same binding environment for ssDNA and protein.

4. Can the authors compare the SWISS-MODEL versus AlphaFold model of *TbRPA1*. Do the conclusions hold?

- Yes, we did compare the models and yes, the conclusions do hold!

The following figure (added in **Supplementary Figure 1d**) shows the comparison of *TbRPA1* DBD-A and DBD-B model structures from SWISS-MODEL and AlphaFold. Since X-ray crystal structures of RPA1 of other species are available, structures from AlphaFold are very similar to homology model structures. Note: since DBD-A and DBD-B are connected by a long loop, the entire geometry of the DBD-A and DBD-B structures from SWISS-MODEL and AlphaFold are different. However, the structures of each domain and the expected binding site of JC-229 are very similar as shown in the structure alignment.

5. In the AlphaFold model, the OB-domains of A and B are very similar. Is there a reason why the authors believe the compound to be specific for OB-A in Tb?

→ In the AlphaFold model in **Supplementary Figure 1e**, the OB-domains of A and B are not very similar as shown below. In addition, the structure alignment of *Tb*DBD-A and *Tb*DBD-B by Pymol software shows poor alignment, implying the amino acid sequences between domains are not homologous. In addition, the SWISS-MODEL structures of *Tb*DBD-A and *Tb*DBD-B are not aligned well by Pymol software. These results can be compared with that of *Hs*RPA1 DBD-F and *Tb*RPA1 DBD-A in **Supplementary Figure 1b, c** (five beta-sheet strands of *Hs*DBD-F and *Tb*DBD-A are well matched with some deviations in the loops, while those of *Hs*DBD-F and *Tb*DBD-B are not). Therefore, *Hs*RPA1 DBD-F is more homologous to *Tb*RPA1 DBD-A than *Tb*RPA1 DBD-B, and there is more chance that JC-229 can bind to *Tb*DBD-A than *Tb*DBD-B.

6. Since the AlphaFold model is a reasonable guide to make truncations, can the authors generate OB-A and OB-B from *Tb*RPA1 and test binding of the compound?

→ We appreciate the reviewer for bringing this up. In human RPA, aromatic residue F238 in DBD-A and W361 in DBD-B are important in ssDNA binding. In human, F238A mutant protein has 1/3 of affinity compared to wild type DBD-AB and W361A has 1/10 of the WT (Walther et al, *Biochemistry*, 1999). F238A W361A double mutant causes >500-fold reduction of the activity, indicating that the presence of either F238 or W361 is sufficient for some binding of *Hs*RPA to ssDNA. Different from human DBD-AB, we observed no binding from *Tb*DBD-AB containing F64A mutation (mutation in *Tb*DBD-A corresponding to F238 in human DBD-A) or *Tb*DBD-AB W188A (mutation in *Tb*DBD-B corresponding to W361 in human DBD-B), suggesting that both DBD-A and B are required for a stable binding of *Tb*DBD-AB with ssDNA (Supplementary Figure 12). To see if this is true, we examined whether *Tb*DBD-A or B alone has any activity in ssDNA binding by EMSA and MST (new data presented in **Supplementary Figure 13**). We purified 6xHis tagged *Tb*DBD-A and *Tb*DBD-B separately. Consistent with *Tb*DBD-AB F64A and W188A mutants' data, *Tb*DBD-A or B alone did not bind ssDNA. These observations suggest that *Tb*DBD-AB works differently from human or yeast DBD-AB proteins. In budding yeast, Yates et al (*Nat Comm*, 2019) showed a high affinity binding of OB-A to ssDNA through MST-based analysis. Similarly, Elias group observed interaction of *T. cruzi* DBD-A to ssDNA in EMSA, but not with *Tc*DBD-B or *Tc*DBD-C (Pavani et al, *PLoS Negl Trop Dis*, 2016). Thus, the ssDNA-binding mode of *Tb*RPA appears to differ from other eukaryotes and even from trypanosomatid RPA with ~80% sequence identities.

7. The biggest concern here is that a compound that supposedly blocks DBD-A is shown to inhibit the ssDNA binding activity of full length *Tb*RPA. In fact, this assumption is incorrect for human RPA. When DBD-A and DBD-B are removed, the trimerization core retains DNA binding activity. In fact, in yeast RPA, the trimerization core provides better stability for ssDNA interactions compared to DBD-A. Thus, the model here predicting that JCC-229 interacts selectively with DBD-A and induces this global inhibition of ssDNA binding by *Tb*RPA does not agree with other recent biochemical investigations.

→ Human and yeast RPA behave very similarly functionally and biochemically, but there are some differences when looking at details (Yates et al, *Nat Commun*, 2018). When it comes to

trypanosomes, they diverged from other model eukaryotes long ago during evolution, so more variations are expected. We demonstrated that *T. brucei* RPA possess ssDNA-binding activity like human and yeast RPA and discovered that the details of binding mode are quite different from the host RPA. We think that it is actually beneficial, because these unique features in *T. brucei* RPA can be exploited without affecting much of the function of host's RPA.

In our study, we tried to understand the biological and biochemical properties of *TbRPA1* and RPA complex and test whether *TbRPA1* can be a good therapeutic target by testing inhibitors or an analog of an inhibitor identified for human RPA1 protein. To do this, we had to generate tools and test assays for the first time for *TbRPA* and we obtained some meaningful data. But unfortunately, we could not cover all aspects of it and investigation of the trimerization core of *TbRPA* is one of those that we couldn't cover in this study. From *in-vitro* ssDNA-binding assay with S105T mutant RPA complex (**Supplementary Figure 18**), we discovered that JC-229 does not inhibit the mutant complex's ssDNA binding (IC_{50} is 33.9 nM for WT vs. ~48,800 nM for S105T mutant complex). One interpretation is that DBD-A of *TbRPA* has the major ssDNA-binding activity and JC-229 targets the DBD-A, thus inhibiting the activity of the full-length complex. Or alternatively, DBD-A has the major dynamic activity of *TbRPA* and the trimerization core has some activity as well. When JC-229 targets the DBD-A, it induces conformational change in *TbRPA* complex, which disrupts the function of the trimerization core as well and thus disrupts *TbRPA* complex's ssDNA binding completely. With S105T mutation, JC-229 may not bind the ssDNA-binding pocket of *TbRPA* and therefore no conformational change could be induced and thus, JC-229 cannot inhibit the mutant protein's ssDNA binding. Crystal or cryoEM data of *TbRPA1* DBD-AB and *TbRPA* complex will be necessary for detailed structural characterization of *TbRPA* protein and to optimize JC-229 further through SAR.

8. However, if the data shown in the paper are taken at face value, another interpretation would be that the compound is binding to multiple domains, or regions that induce allosteric changes in *TbRPA1*. Given that most of the interaction data is speculative or derived from investigation of either DBD-A or DBD-AB, the authors should be a bit more circumspect in their interpretation.

→ Comment 8 is related to 6 and 7, so we responded in 6 and 7. We revised the text more carefully.

Minor Comments:

1. Abstract line 1: complex is repeated twice.

→ Corrected.

2. Line 90: The statement that there are 6 DBDs in RPA is incorrect. There are 6 OB domains and four of these primarily coordinate DNA binding and are called DBDs.

→ Thank you for the suggestion. The terminology "DBD" for all RPA domains have been widely used in literatures so we did not change the terminology. However, as suggested, we have rephrased the sentence as such: "Eukaryotic RPA subunits, including the human ortholog, contain 6 oligosaccharide/oligonucleotide-binding (OB) folds: four in the largest subunit RPA1 and one each in RPA2 and RPA3¹⁴. The OB fold is found in many proteins with ssDNA-binding functions^{21,22}. RPA1 has domains called DBD (DNA-binding domain): DBD-F, A, B, and C in tandem. Each of these DBDs contains an OB-fold structure. While DBD-F also contains an OB-fold structure, it is involved in mediation of protein-protein interactions (PPIs). The central region of the RPA1, DBD-A and B (DBD-AB), is involved in dynamic interaction with ssDNA and DBD-C is required for the interaction of RPA1 with RPA2^{15,22-24}." (page 4, lines 89-96).

3. Line 94: The statement that DBDs A and B ‘dominate’ during ssDNA is being revised through new studies looking at the dynamics of RPA domains. Please refer to more current literature on RPA dynamics.

→ We have referred to a new citation (Pokhrel et al, *Nat Struct Mol Biol*, 2019).

4. Line 111: Shouldn't drug discovery follow a thorough characterization of the target enzyme? How can you assess drug efficacy and mechanism of action without knowledge of the enzyme in question.

→ We agree that characterization of the target enzyme is required to assess drug efficiency and mode of action. In this study, while focusing on the genetic and biochemical characterization of *TbRPA*, we decided to test some of the inhibitors for their toxic effects on *T. brucei* cells as well. Identification of inhibitors is important in therapy development, but it is also important to study the function of a target protein. These inhibitors are particularly useful for genetic interaction studies, such as synthetic lethal interaction. Through this revision, we have had the chance to further characterize *TbRPA* protein and we hope that additional experiments that we've done for the revision are satisfactory to the reviewers and addressed the reviewers' concerns.

5. Line 114: DBD-A of eukaryotic RPA is the major ssDNA binding domain. This statement is inaccurate. DBDs A and B are more dynamic compared to the Trimerization core of RPA.

→ In line 114, we wrote “the DBD-A of *TcRPA1* is the major ssDNA-binding domain, similar to other eukaryotes”. We have rephrased the sentence to “In *T. cruzi*, electrophoretic mobility shift assay (EMSA) using recombinant DBD-A, B, or C demonstrated that *TcRPA1* DBD-A has the strongest ssDNA-binding activity.” (page 5, lines 112-114)

6. Line 125: There is so many contradictions in the introduction and the logic for this paper. The authors state that the drug discovery effort provides a unique opportunity as the *Tb* RPA does not possess a F-domain. However, they are targeting DBD-A. This is present in human RPA as well. How does their approach involve an advantage by targeting DBD-A?

→ We have removed “the lack of DBD-F in RPA1” (page 5, lines 123-125).

DBD-F has an OB fold structure and some inhibitors bind OB fold. So, it is possible that these inhibitors can bind other DBDs as they, including DBD-A, have an OB-fold. For this reason, generally, DBD-F inhibitors are also tested for inhibition of ssDNA binding of DBD-AB, because those that inhibit ssDNA-binding activity are not desirable ones sometimes in cancer therapy [Glanzer et al, *Cancer Res*, 2014, Frank et al, *J Med Chem*, 2013]. In our study which aims to validate a potential target of chemical probes, we looked for those inhibitors that are specifically active on the inhibition of ssDNA-binding activity of *Tb*, but not that of *HsRPA*. These were not clearly explained in our initial manuscript as reviewer 1 also pointed it out in his/her concern 6, so we thoroughly revised the manuscript to avoid any confusion.

7. Line 131: What is an in-house 3D structural model? Just state SWISS-MODEL or use the new AlphaFold Model (which looks quite reasonable)

→ We removed “in-house 3D structural model” in that sentence. In another place, “3D model” has been replaced with SWISS-MODEL (please see page 6, lines 155-162).

8. A 2 to 5-fold difference in affinity is not an ideal drug target.

→ We are not exactly sure on what aspect the reviewer brought this up. We assumed it could be this part (line 320 in the first submission), “*TbRPA1* DBD-AB has a weaker binding affinity to ssDNA compared to the *TbRPA* complex (~2 fold), but a much higher concentration of JC-229 is needed

to inhibit the activity of *Tb*DBD-AB than *Tb*RPA complex (228 nM vs. 24.5 nM).” If this is what the reviewer is mentioning, we are comparing the numbers obtained for *Tb*RPA and *Tb*DBD-AB, but not claiming that it is an ideal drug target. If this sentence is not the one, please let us know in which sentence you have concerns.

9. *Line 383: ‘Binding pose’ is not the correct terminology*

- ➔ We used the term “pose” because it means “position and orientation in 3D”, while position may mean more of “location”. We changed the term “pose” to “position or orientation”, as suggested by the reviewer.

REVIEWERS' COMMENTS

Reviewer #1 (Remarks to the Author):

The revised manuscript is much improved and the additional experiments have addressed my major concerns.

Minor

The statement that "RPA protects genome integrity..." is awkward. I'm not sure what "protection" means in this context

Line 328 states "If more JC-229 is required to inhibit TbDBD-AB than the TbRPA complex because dT32 can harbor more molecules of TbDBD-AB..." state a reason for the possible difference but it is not the only one. Additional interactions on longer DNA substrates and hence inherent lower affinity for shorter substrates would result in less inhibitor being necessary, if true equilibrium conditions are achieved.

Reviewer #3 (Remarks to the Author):

The authors have done an excellent and comprehensive job of addressing all of my concerns.

Response to reviewers (second revision)

We are very much delighted that reviewers were satisfied with our revised version. Please see below our response to the reviewer 1's comments/concerns.

In the second revision (main text and supplementary information files), revised text according to the reviewer's comments are shown in blue font.

Please see below our point-by-point response to reviewer 1's comments/concerns.

REVIEWERS' COMMENTS

Reviewer #1 (Remarks to the Author):

The revised manuscript is much improved and the additional experiments have addressed my major concerns.

→ Thank you very much!

Minor

The statement that "RPA protects genome integrity..." is awkward. I'm not sure what "protection" means in this context

→ We revised the sentence to "RPA protects the exposed single-stranded DNA (ssDNA) during DNA replication and repair." In line 21-22 (Abstract).

Line 328 states "If more JC-229 is required to inhibit TbDBD-AB than the TbRPA complex because dT₃₂ can harbor more molecules of TbDBD-AB..." state a reason for the possible difference but it is not the only one. Additional interactions on longer DNA substrates and hence inherent lower affinity for shorter substrates would result in less inhibitor being necessary, if true equilibrium conditions are achieved.

→ We also expected what the reviewer described in "Additional interactions on longer DNA substrates and hence inherent lower affinity for shorter substrates would result in less inhibitor being necessary". But we observed the opposite. Interaction with dT₃₂ and dT₂₀ for *Tb*DBD-AB was similar (K_d : 30.5 nM for dT₂₀ vs. 24.6 nM for dT₃₂), but the amount of JC-229 required for shorter oligo (dT₂₀) was much higher. We do not have a clear explanation for this result at this moment. One possibility is that there might be some interaction between *Tb*DBD-AB molecules. When competing with JC-229, this interaction may make shorter oligo dT₂₀ compete better than dT₃₂. Crystal structure could help better understand the binding mode of *Tb*RPA with ssDNA and JC-229.

Reviewer #3 (Remarks to the Author):

The authors have done an excellent and comprehensive job of addressing all of my concerns.

→ Thank you very much!!